

# Glacier shrinkage in the Alps continues unabated as revealed by a new glacier inventory from Sentinel-2

*Frank Paul[1], Philipp Rastner[1], Roberto Sergio Azzoni[2], Guglielmina Diolaiuti[2], Davide Fugazza[2], Raymond Le Bris[1], Johanna Nemec[3], Antoine Rabatel[4], Mélanie Ramusovic[4], Gabriele Schwaizer[3], Claudio Smiraglia[2]*

Department of Geography, University of Zurich, Zurich, Switzerland
Department of Environmental Science and Policy, University of Milan, Milan, Italy
ENVEO IT GmbH, Innsbruck, Austria
Univ. Grenoble Alpes, CNRS, IRD, Grenoble-INP, Institut des Géosciences de l'Environnement (IGE, UMR5001), Grenoble, France

*Correspondence: Frank Paul (frank.paul@geo.uzh.ch)*

## Abstract

The on-going glacier shrinkage in the Alps requires frequent updates of glacier outlines to provide an accurate database for monitoring or modeling purposes (e.g. determination of run-off, mass balance, or future glacier extent) and other applications. With the launch of the first Sentinel-2 (S2) satellite in 2015, it became possible to create a consistent, Alpine-wide glacier inventory with an unprecedented spatial resolution of 10 m. Fortunately, already the first S2 images acquired in August 2015 provided excellent mapping conditions for most of the glacierised regions in the Alps. We have used this opportunity to compile a new Alpine-wide glacier inventory in a collaborative team effort. In all countries, glacier outlines from the latest national inventories have been used as a guide to compile a consistent update. However, cloud cover over many glaciers in Italy required including also S2 scenes from 2016. Whereas the automated mapping of clean glacier ice was straightforward using the band ratio method, the numerous debris-covered glaciers required intense manual editing. The uncertainty in the outlines was determined with multiple digitising of 14 glaciers by all participants. Topographic information for all glaciers was derived from the ALOS AW3D30 DEM. Overall, we derived a total glacier area of 1806 ±60 km$^2$ when considering 4394 glaciers >0.01 km$^2$. This is 14% (-1.2%/a) less than the 2100 km$^2$ derived from Landsat scenes acquired in 2003 and indicating an unabated continuation of glacier shrinkage in the Alps since the mid-1980s. Due to the higher spatial resolution of S2 many small glaciers were additionally mapped in the new inventory or increased in size compared to 2003. An artificial reduction to the former extents would thus result in an even higher overall area loss. Still, the uncertainty assessment revealed locally considerable differences in interpretation of debris-covered glaciers, resulting in limitations for change assessment when using glacier extents digitised by different analysts. The inventory is available at: doi.pangaea.de/10.1594/PANGAEA.909133 (Paul et al., 2019).




## 1. Introduction

Precise information on glacier extents is required for numerous glaciological and hydrological cal-
culations, ranging from the determination of glacier volume, surface mass balance and future glac-
ier evolution to run-off, hydro-power production, and sea-level rise (e.g., Marzeion et al., 2017).
For these and several other applications glacier outlines spatially constrain all calculations thus
providing an important baseline dataset. In response to the on-going atmospheric warming, glaci-
ers retreat, shrink and lose mass in most regions of the world (e.g., Gardner et al. 2013, Wouters et
al. 2019, Zemp et al. 2019). Accordingly, a frequent update of glacier inventories is required to
reduce uncertainties in subsequent calculations. With relative area loss rates of about 1% per year
in many regions globally (Vaughan et al. 2013), glaciers lose about 10% of their area within a dec-
ade and a decadal update frequency seems sensible. In regions with stronger glacier shrinkage such
as the tropical Andes (e.g. Rabatel et al. 2013, 2018) or the European Alps (e.g. Gardent et al.
2014) an even higher update frequency is likely required. However, apart from the high workload
required to digitise or manually correct glacier outlines (e.g. Racoviteanu et al. 2009), it is often
not possible to obtain satellite images in a desired period of the year with appropriate mapping
conditions, i.e. without seasonal snow and clouds hiding glaciers. Hence, glacier inventories are
often compiled from images acquired over several years resulting in a temporarily inhomogeneous
dataset. Fortunately, a 3-year period of acquisition is still acceptable in error terms, as area chang-
es of about ±2% are within the typical area uncertainty of about 3 to 5% (e.g. Paul et al. 2013).

The last glacier inventory covering the entire Alps with a common and homogeneous date has
been compiled from Landsat Thematic Mapper (TM) images acquired within six weeks in the
summer of 2003 (Paul et al. 2011). Although this dataset has its caveats (e.g. missing small glaci-
ers in Italy and some debris-covered ice), it is methodologically and temporarily consistent and
representing glacier outlines of the Alps in the Randolph Glacier Inventory (RGI). A few years
later, high quality glacier inventories were compiled from very high-resolution datasets (aerial
photography, airborne laser scanning) on a national level in all four countries of the Alps with sub-
stantial glacier coverage (Austria, France, Italy, Switzerland). These more recent inventories refer
to the periods 2008-2011 for Switzerland (Fischer et al. 2014), 2004-2011 for Austria (Fischer et
al. 2015), 2006-2009 for France (Gardent et al. 2014), and 2005-2011 for Italy (Smiraglia et al.
2015). As an 8-year period is rather long, consistent and comparable change assessment is chal-
lenging. However, for the first version of the World Glacier Inventory (WGI) the temporal spread
was even larger, ranging from 1959 to about 1983 (Zemp et al. 2008). Another problem for change
assessment is the inhomogenous interpretation of glacier extents, in part to be compliant with the
analysis in earlier inventories. Hence, calculations over the entire Alps that require a consistent
time stamp are difficult to perform and rates of glacier change are difficult to compare across re-
gions (e.g. Gardent et al. 2014).






Considering the on-going strong glacier shrinkage in the Alps over the past decades and the above
shortcomings of existing datasets, there is a high demand to compile a (1) new, (2) precise and (3)
consistent glacier inventory for the entire Alps, with data acquired under (4) good mapping condi-
tions in (5) a single year. Although it might be difficult to satisfy all five criteria at the same time,
at least some of them seem achievable by means of recently available satellite data. With the 10 m
resolution data from Sentinel-2 (S2) and its 290 km swath width it is possible (a) to improve the
quality of the derived glacier outlines (compared to Landsat TM) substantially (Paul et al. 2016)
and (b) to cover a region such as the Alps with a few scenes acquired within a few weeks or even
days, satisfying criteria (2) and (5). Good mapping conditions, however, only occur by chance af-
ter a comparably warm summer when all seasonal snow off glaciers has melted and largely cloud
free conditions persist over an extended time span in August or September.

In this study we present a new glacier inventory for the European Alps that has been compiled
from S2 data that were mostly acquired within two weeks of August 2015 (during the commission-
ing phase). However, due to glaciers (mostly in in Italy) being partly cloud-covered, also scenes
from 2016 (and very few from 2017) were used. Hence, criterion (5) could not be fully satisfied. In
order to satisfy point (3), we decided to perform the mapping of clean ice with an identical method
(band ratio), and distribute the raw outlines to the national experts for editing of wrongly classified
regions (e.g. adding missing ice in shadow and under local clouds or debris cover, removing lakes
and other water surfaces). As a guide for the interpretation the analysts used the latest high-
resolution inventory in each country. All corrected datasets were merged into one dataset and
topographic information for each glacier was derived from the ALOS AW3D30 DEM. For uncer-
tainty assessment all participants corrected the extents of 14 glaciers independently four times.

## 2. Study region
The Alps are a largely west-east (south-north in the West) oriented mountain range in the centre of
Europe (roughly from 2° to 18° E and 43° to 49° N) with peaks reaching 4808 m a.s.l. in the West
at Mt. Blanc/Monte Bianco and elevations above 3000 m a.s.l. in most regions. In Fig. 1 we show
the region covered by glaciers along with footprints of the tiles used for data processing. The Alps
act thus as a topographic barrier for air masses coming from the North and South as well as from
the West in the western part. This results in enhanced orographic precipitation and a high regional
variability of precipitation amounts in specific years as well as in the long-term mean (e.g. Frei et
al. 2003). On the other hand, temperatures are horizontally rather uniform (e.g. Böhm et al. 2001)
but vary strongly with height according to the atmospheric lapse rate (e.g. Frei 2014). Snow accu-
mulation is mostly due to winter precipitation, but some snowfall can also occur in summer at
higher elevations, reducing ablation for a few days.



There is no significant long-term trend in precipitation over the last 100+ years (Casty et al. 2005),
but summer temperatures in the Alps have increased sharply (by about 1 °C) in the mid-1980s (e.g.
Beniston 1997, Reid et al. 2016). In consequence, winter snow cover barely survives the summer
even at high elevations and / or when strong positive deviations in temperature occurred. Glacier
mass balances in the Alps were thus pre-dominantly negative over the past three decades (e.g.
Zemp et al. 2015) and the related mass loss resulted in widespread glacier shrinkage and disinte-
gration over the past decades (e.g. Gardent et al. 2014, Paul et al. 2004). With a total area of about
2000 km$^2$ in 2003 and a mean annual mass loss of about 1 m w.e. per year, the European Alps cur-
rently lose about 2 Gt of ice per year.

Most glaciers in the Alps are of cirque, mountain and valley type and the two largest ones (Aletsch
and Gorner glaciers) have an area of about 80 km$^2$ and 60 km$^2$, respectively. Some glaciers reach
down to 1300 m a.s.l., and the overall mean elevation is around 3000 m a.s.l., a unique value com-
pared to other regions of the RGI (e.g. Pfeffer et al. 2014). Due to the surrounding often ice-free
rock walls of considerable height, many glaciers in the Alps are heavily debris-covered. Whereas
this allowed the tongues of several large valley glaciers to survive at comparably low elevations
(Mölg et al. 2019), many glaciers - large and small - become invisible under increasing amounts of
debris. Combined with the on-going down-wasting and disintegration, precisely mapping their ex-
tents is increasingly challenging.
*Figure 1*

## 3. Datasets
### 3.1 Satellite data
In total, 23 S2 tiles were processed to cover the study region with cloud free images (Figure 1 and
Table 1). Of these, 11 were acquired in 2015, 9 in 2016 and 3 in 2017. Convective clouds in Italy
(mostly along the Alpine main divide) required stretching the main acquisition period over two
years. All glaciers in France were mapped from four tiles acquired on 29.8.2015. This date covers
also most glaciers mapped in Switzerland (five tiles) apart from the south-east tile 32TNS that was
acquired three days earlier (26.8.2015). Three tiles from that date (32TNT / TNS / TPT) are used
to map glaciers in western-Austria and three tiles (32TQT / 32TQS / 33TUN) from 27.8.2016 for
the eastern part of Austria. Twelve tiles had to be used to map glaciers in Italy of which two are
from 2015, seven are from 2016, and three from 2017 (Fig. 1). However, the latter three only cov-
er very few and small glaciers so that collectively the northern (Switzerland / Austria) and western
(France) parts of the inventory are from 2015 whereas the southern (Italy) and eastern (Austria)
parts are from 2016. All tiles were downloaded from remotepixel.ca (only the required bands, no
longer possible), earthexplorer.usgs.gov or the Copernicus Open Access Hub.



*Table 1*
From all tiles, bands 2, 3, 4, 8, and 11 (blue, green, red, Near Infra-Red / NIR, Short Wave Infra-
Red / SWIR) of the sensor Multi Spectral Imager (MSI) were downloaded and colour composites
were created from the 10 m visible and NIR (VNIR) bands. The 20 m SWIR band 11 was bilinear-
ly resampled to 10 m resolution to obtain glacier outlines at this resolution. The 10 m resolution
VNIR bands allowed for a much better identification of glacier extents (e.g. correcting debris-
covered parts) than possible with Landsat (Paul et al. 2016), resulting in a higher quality of the
outlines. Apart from the resampling, all image bands are used as they are except for Austria, where
further pre-processing has been applied (see Section 4.2.1). The August 2015 scenes from the S2
commissioning phase had reflectance values stretched from 1 to 1000 (12 bit) instead of the later
16 bit (allowing values up to 65536), but this linear rescaling had no impact on the threshold value
for the band ratio (see Section 4.1).

**3.2 Digital elevation models (DEMs)**
We originally intended using the new TanDEM-X (TDX) DEM to derive topographic information
for all glaciers, as it covers the entire Alps and was acquired closest (around 2013) to the satellite
images used to create the inventory. However, closer inspection revealed that it had data voids and
suffered from severe artefacts (Fig. 2). Although these are mostly located in the steep terrain out-
side of glaciers, many smaller glaciers are severely impacted, resulting in wrong topographic in-
formation. As an alternative we investigated the ALOS AW3D30 DEM that was compiled from
ALOS tri-stereo scenes (Takaku et al. 2014) and acquired about five years before the TDX DEM
(around 2008). The AW3D30 DEM has a less good temporal match but no data voids and compa-
rably few artefacts (Fig. 2). The individual tiles were merged into one 30 m dataset in UTM 32N
projection with WGS84 datum. For the pre-processing of satellite bands in Austria, a national
DEM with 10 m resolution derived from laser scanning was used (Open Data Österreich: da-
ta.gv.at).
*Figure 2*
**3.3 Previous glacier inventories**
As mentioned above, outlines from previous national glacier inventories were used to guide the
delineation. They have been mostly compiled from aerial photography with very high spatial reso-
lution (better than 1 m) and should thus provide the highest possible quality. This allowed consid-
ering very small and otherwise unnoticed glaciers and helped to identify glacier zones that are de-
bris covered. The substantial glacier retreat that took place between the two inventories was well
visible in most cases and did not hamper the interpretation. However, a larger number of very
small glaciers were not mapped in 2003 and have now been added or digitised with larger extents.
A large issue with respect to additional work load is the compilation of ice divides. They can be
derived semi-automatically from watershed analysis of a DEM using a range of methods (e.g.
Kienholz et al. 2013), but in general numerous manual corrections have still to be applied. To pro-
vide some consistency with previous national inventories, we decided using the drainage divides
from these inventories to separate glacier complexes into entities. However, due to the locally poor
geolocation of the S2 scenes (Kääb et al. 2016, Stumpf et al. 2018), the location of the ice divides
was partly manually adjusted.

## 198    4. Methods

### 199    4.1 Mapping of clean ice in all regions

Automated mapping of clean to slightly dirty glacier ice is straight forward using a red or NIR to
SWIR band ratio and a (manually selected) threshold (e.g. Paul et al. 2002). Also other methods
such as the normalised difference snow index (NDSI) work well (e.g. Racoviteanu et al. 2009) as
both utilise the strong difference in reflectance from the VNIR to the SWIR for snow and ice (e.g.
Dozier 1989). As the latter are bright in the VNIR bands (high reflectance) but very dark (low re-
flectance) in the SWIR, dividing a VNIR band by a SWIR band gives high values over glacier ice
and snow and very low ones over all other terrain as this is often much brighter in the SWIR than
the VNIR. The manual selection of a threshold for each scene (or S2 tile) has the advantage to in-
clude a regional adjustment of the threshold to local atmospheric conditions. We followed the rec-
ommendation to select the threshold in a way that good mapping results in regions with shadow
are achieved. By lowering the threshold, more and more bare rock in shadow is included, creating
a very noisy result. It has been shown in a previous study (Paul et al. 2016) that glacier mapping
with S2 (using a red / SWIR ratio) requires an additional threshold in the blue band to remove
misclassified rock in shadow. Hence, for this inventory glaciers have been first automatically iden-
tified following the equation:

(red / SWIR) > *th1* and blue > *th2*


with the empirically derived thresholds *th1* and *th2*. As mentioned above, the SWIR band was bi-
linearly resampled from 20 to 10 m spatial resolution before computing the ratio. No filter for im-
age smoothing was applied to retain fine spatial details, such as rock outcrops. Figure 3 shows for
a test site in the Mt. Blanc region (Leschaux Glacier) the impact of the threshold selection. Figure
3a depicts the (contrast stretched) red / SWIR ratio image, Fig. 3b the impact of *th1* on the mapped
area, Fig. 3c the impact of *th2*, and Fig. 3d the resulting outlines after raster-vector conversion. As
can be seen in Fig. 3b, there is very little impact on the mapped glacier area when increasing *th1* in
steps of 0.2. For this region we used 3.0 as t*h1* resulting in the blue and yellow areas as the
mapped glacier. Wrongly mapped rock in shadow is then reduced back with *th2* (Fig. 3c). In this
case a value of 860 was selected for *th2* i.e. only the blue area is considered. This correctly re-





moved rock in shadow from the glacier mask for the region to the right of the white arrow but, on
the other hand, correctly mapped ice in shadow is removed at the same time in the region above
the green arrow (Figs. 3c and d). Hence, threshold selection is always a compromise as it is in gen-
eral not possible to map everything correctly with one set of thresholds. The resulting glacier maps
for all regions were converted to a shape file using raster-vector conversion and by setting the non-
glacier class to '*no data*' before. In the resulting shape file internal rocks are thus data voids.
All pre-processed scenes were provided in their original geometry for correction by the national
experts. As shown in Fig. 3c, it was sometimes not possible to include dark bare ice and at the
same time exclude bare rock in shadow. Such wrongly classified regions were corrected by the
analysts together with data gaps for debris cover and clouds (omission errors), wrongly mapped
water bodies (e.g. turbid lakes and rivers) and shadow regions (commission errors). By setting the
minimum glaciers size to 0.01 km$^2$, most of the often very small snow patches were removed.
*Figure 3*

## 4.2 Corrections in the different countries

### 4.2.1 Austria

The satellite scenes for Austria were further pre-processed (see Paul et al. 2016) to remove water
surfaces and improve classification of glacier ice in cast shadow, before manual corrections were
applied. The latter work was mainly performed by one person (J. Nemec). Two previous Austrian
glacier inventories (Lambrecht and Kuhn 2007, Fischer et al. 2015) were used to support the inter-
pretation of small glaciers, debris covered glacier parts, and the boundary across common accumu-
lation areas. Further, an internal independent quality control of the generated glacier outlines was
made by a second person (G. Schwaizer), using orthophotos (30 cm pixel spacing) acquired in late
August 2015 for most Austrian glaciers for overall accuracy checks and to assure the correct delin-
eation of debris covered glacier areas. In Fig. 4a we illustrate the strong glacier shrinkage from
1998 (yellow lines) to 2016 (red) as well as the manual corrections applied, extending the bright
filled areas of the raw classification to the red extents.

### 4.2.2 France

The raw glacier outlines from S2 were corrected by one person (A. Rabatel). The glacier outlines
from the previous inventory by Gardent et al. (2014) were used for the interpretation, in particular
in shadow regions and for glaciers under debris cover. It is noteworthy that the previous inventory
was made on the basis of aerial photographs (2006-2009) with field campaigns for the debris-
covered glacier tongues to clarify the outline delineation. As a consequence, this previous invento-
ry constitutes a highly valuable reference. In addition, because even on debris-covered glaciers the
changes between 2006-09 and 2015 are important (Fig. 4b), Pléiades images from 2015-2016 ac-
quired within the KALIDEOS-Alpes / CNES program were use as a guideline, mostly for the



heavily debris-covered glacier tongues.

**4.2.3 Italy**
The raw glacier outlines from S2 were corrected by two analysts (D. Fugazza, R.S. Azzoni). The
outlines were separated into regions based on the administrative division of Italy, following the
previous Italian glacier inventory (Smiraglia et al. 2015). From west to east, the regions are Aosta
Valley, Piemonte, Lombardy, Trento Province, Bolzano Province, Veneto, Friuli, Venezia Giulia.

As mentioned in the Introduction, clouds covered the southern Alpine sector on the S2 scenes from
August 2015. Hence, most of the inventory was compiled based on images from August and Sep-
tember 2016 and 3 scenes from 2017 (one in August and two in October) were also used to map
glaciers that were under clouds or with adverse mapping conditions, i.e. excessive snow cover or
shadows in the other scenes. Images acquired in August had little residual seasonal snow and a
high solar elevation at the time of acquisition, which minimised shadow areas creating very good
mapping conditions. In September 2016 and October 2017, more snow was present on high moun-
tain cirques and glacier tongues, but comparatively few snow patches were found outside glaciers.
However, the lower solar elevation compared to August caused a few north-facing glaciers and
glacier accumulation areas to be under shadows.

Seasonal snow and rocks in shadow that were wrongly identified as clean ice were manually delet-
ed by the analysts, as well as lakes and large rivers. In shadow regions, and for glaciers with large
debris cover, the outlines from the previous Italian inventory by Smiraglia et al. (2015) were par-
ticularly valuable as a guide. Where glaciers were entirely under shadows, the outlines from the
previous inventory were copied without changes, while in case of partial shadow coverage they
were edited in their visible portions.

Glaciers in three sectors of the Alps, i.e. the Orobie Alps, Dolomites and Julian Alps posed signifi-
cant challenges for mapping. The three regions host very small niche glaciers and glacierets: in the
Orobie and Julian Alps, their survival is granted by abundant snow-falls, northerly aspect and ac-
cumulation from avalanches, with debris cover also playing an important role. In the Dolomites,
debris cover is often complete (Smiraglia and Diolaiuti 2015), while the steep rock walls provide
shadow and further complicate mapping.

For glaciers in the Orobie Alps, an aerial orthophoto acquired by Regione Lombardia (geopor-
tale.regione.lombardia.it) in 2015 was used to aid the interpretation in view of its finer spatial reso-
lution, although the image also shows evidence of seasonal snow. Here, manual delineation of the
glacier outlines was required as the band ratio approach could only detect small snow patches (see
Fig. 4c). In the other two regions, outlines from the previous inventory, derived from aerial ortho-



photos acquired in 2011, were copied and only corrected where evidence of glacier retreat was
found. While the uncertainty in the outlines of these latter glaciers is likely large, the combined
glacier area from the three regions is just above 1% of the total area of Italian glaciers.

**4.2.4 Switzerland**
The raw glacier outlines from S2 were corrected by three persons (R. LeBris, F. Paul, P. Rastner)
each of them being responsible for a different main region (south of Rhone, north of Rhone/Rhine,
south of Rhine). The glacier outlines from the previous inventory by Fischer et al. (2014) were
highly valuable for the interpretation, in particular in shadow regions and for glaciers under debris
cover. In the hot summer of 2015 most seasonal snow had disappeared by the end of August so
that mapping conditions with a comparably high solar elevation (limited regions in shadow) were
very good. Some glaciers that could not be identified in the (contrast-stretched) S2 images were
either copied from the previous inventory (if located in shadow) or assumed to have disappeared
(if sun-lit). Wrongly mapped (turbid) lakes and rivers (Rhone, Aare) were manually removed.

In a few cases (mostly debris-covered glaciers) we had to deviate from the interpretation of the
previous inventories. As shown in Fig. 4d, very high-resolution satellite imagery (as sometimes
available in Google Earth) or aerial photography do not always help for a 'correct' interpretation
of glacier extents, as the rules applied for identification of ice under debris cover might differ. In
this case it seems that the debris-covered region was not corrected in the 2003 and 2008 invento-
ries, but is now largely included. This glacier has thus strongly grown since 2003 due to a new in-
terpretation and the better visibility of debris cover with S2.
*Figure 4*

**4.3. Drainage divides and topographic information**
Drainage divides between glaciers were copied from previous national inventories but were locally
adjusted along national boundaries. In part this was required because different DEMs had been
used in each country to determine the location of the divide. Additionally, some glaciers are divid-
ed by national boundaries rather than flow divides. This can result in an arbitrary part of the glaci-
er (e.g. its accumulation zone) being located in one country and the other part (e.g. its ablation
zone) in another country. As this makes no sense from a glaciological (and hydrological) point of
view, such glaciers (e.g. Hochjochferner in the Ötztal Alps) have been corrected in a way that they
belong to the country where the terminus is located. There are thus a few inconsistencies in this
inventory compared to the national ones.

After digital intersection of glacier outlines with drainage divides, topographic information for
each glacier entity is calculated from both DEMs (ALOS and TDX) following Paul et al. (2009).
The calculation is fully automated and applies the concept of zone statistics introduced by Paul et



al. (2002). Each region with a common ID (this includes regenerated glaciers consisting of two
polygons) is interpreted as a zone over which statistical information (e.g. minimum / maximum /
mean elevation) is derived from an underlying value grid (e.g. a DEM or a DEM-derived slope and
aspect grid). Apart from glacier area (in km$^2$) all glaciers have information about mean, median,
maximum and minimum elevations, mean slope and aspect (both in degrees) and aspect sector
(eight cardinal directions) using letters and numbers (N=1, NE=2, etc.). Further information ap-
pended to each glacier in the attribute table of the shape file is the satellite tile used, the acquisition
date, the analyst and the funding source. This information is applied automatically by digital inter-
section ('*spatial join*') to all glaciers from a manually corrected scene footprint shape file (see Fig.
1). The various attributes have then been used for displaying key characteristics of the datasets in
bar graphs, scatter plots and maps (see Section 5.1).

## 4.4 Change assessment

Glacier area changes have only been calculated with respect to the inventory from 2003, as the
dates for the previous national inventories were too diverse for a meaningful assessment (see In-
troduction). To obtain consistent changes, only glaciers that are also mapped in the 2003 inventory
are used for a direct comparison (automatically selected via a '*point in polygon*' check). After real-
ising that a glacier-specific comparison is not possible due to differences in interpretation, we de-
cided to only compare the total glacier area of the previous and new inventory.

## 4.5 Uncertainty assessment

As several analysts have digitised the new inventory, we decided performing multiple digitising of
a pre-selected set of glaciers to determine internal variability in interpretation per participant and
across participants as a measure of the uncertainty of the generated dataset. For this purpose, all
participants used the same raw outlines from S2 tile 32TLR to manually correct 14 glaciers (sizes
from 0.1 to 10 km$^2$) to the south of Lac des Dix around Mt. Blanc de Cheilon (3870 m a.s.l.) for
debris cover. All glaciers were digitised 4 times by 5 participants giving a nominal total of 280
outlines for comparison. Results were analysed using an overlay of outlines to identify the general
deviations in interpretation and through a glacier-by-glacier comparison of glacier sizes. For the
latter all datasets were intersected with the same drainage divides and glacier-specific areas were
calculated. For each glacier and the entire region, mean area values and standard deviations are
calculated per glacier, per participant and for the total sample. The participants were asked to only
use the S2 image and the 2003 outlines as a guide for interpretation in the first two digitisation
rounds and consider interpretation of very high-resolution imagery as provided by Google Earth
for the second two rounds. At a minimum, one day should have passed between each digitisation
round and it was not allowed to show any of the former outlines. On average, each digitisation
round took about 2 hours.

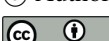



Additionally, we applied the buffer method to obtain a statistical uncertainty value for the entire
sample. This method gives a minimum and maximum area and was used to determine a relative
area difference. This value multiplied by 0.68 gives the standard deviation (assuming normally
distributed deviations from the correct outline) that is used as a further measure of area uncertainty
(Paul et al. 2017). The selected buffer is based on an earlier multiple digitising experiment for a
couple of glaciers (Paul et al. 2013) showing that the variability in the positioning is within one
pixel (or about ±10 m in the current case) to both sides of the 'true' vector line. Strictly, a larger
buffer should be used for the debris-covered glacier parts, as their uncertainty is higher. However,
we have not implemented this here, as the related calculations are computationally expensive (cf.
Mölg et al. 2018). Instead, we also applied a ±2 pixels buffer to all glaciers. Depending on the de-
gree of debris coverage, a realistic uncertainty estimate is likely between these two values.

## 394 5. Results

### 395 5.1 The new glacier inventory

In total, we identified 4394 glaciers larger than 0.01 km$^2$ covering a total area of 1805.77 km$^2$, of
which 361.5 km$^2$ (20%) is found in Austria and 227.1 (12.6%), 325.3 (18%), and 892.0 km$^2$
(49.4%) in France, Italy, and Switzerland, respectively. The size class distribution by area and
count is depicted in Fig. 5a and also listed in Table 2. In total, 63% (92%) of all glaciers are small-
er than 0.1 km$^2$ (1.0 km$^2$) covering 5.5% (28%) of the glacierised area, whereas 1.6% are larger
than 5 km$^2$ and cover 40%. Thereby, glaciers in the size class 1 to 5 km$^2$ alone cover one third
(32%) of the area but only 6% of the total number. This biased size class distribution is typical for
alpine glaciers where a few large glaciers are surrounded by numerous much smaller ones. The
distribution of glacier number and area by aspect sector displayed in Fig. 5b shows the dominance,
both in number and coverage area, of northerly exposed glaciers compared to all other sectors.
About 60% of all glaciers (covering 60% of the area) are exposed to the NW, N, or NE whereas
only 21% of all glaciers are found in the sectors SE, S, and SW. This distribution of glacier aspects
is typical for regions where radiation plays a larger role in glacier existence compared to factors
such as precipitation (Evans and Cox, 2005). The larger area coverage for glaciers facing SE is
mostly due to the large Aletsch and Fiescher glaciers.

*Figure 5, Table 2*


A plot of glacier surface area *vs.* minimum and maximum elevations (Fig. 6a) reveals that glaciers
smaller than 1 km$^2$ can be found at all elevations, indicating that their mean elevation does only
slightly depend on climatic factors. Glaciers larger than 1 km$^2$ on the other hand have clearly dis-
tinguished maximum and minimum elevations, *i.e.* they arrange around a climatically driven mean
elevation which is around 3000 m a.s.l. Plotting glacier area *vs.* elevation range (Fig. 6b) shows
that the largest glaciers are not those with the highest elevation range (the maximum of 3140 m is





for Glacier des Bossons in the Mont-Blanc massif with a size of 10 km$^2$) and that for the majority
of glaciers the elevation range increases with glacier size. This is typical for regions dominated by
mountain and valley glaciers as these follow the given topography. The ca. 7 km$^2$ large Plaine
Morte Glacier is a plateau glacier with an elevation range of only 350 m and represents an excep-
tion from the rule.
*Figure 6*

The median elevation of a glacier is largely driven by temperature, precipitation and radiation. As
temperature is rather similar at the same elevation over large regions and topography (aspect /
shading) has a strong local impact on radiation receipt, the large-scale variability of median (or
mean) elevation of a glacier has a high correlation with precipitation amounts (e.g. Ohmura et al.
1992, Oerlemans 2005, Rastner et al. 2012, Sakai et al. 2015). The spatial distribution of glacier
median elevations in the Alps (Fig. 7a) thus also reflects the general pattern of annual precipitation
amounts (e.g. Frei et al. 2003). When focusing on glaciers larger than 0.5 km$^2$ (that are less im-
pacted by local topographic conditions), clearly lower median elevations (around 2400 m a.s.l.) are
found for glaciers along the northern margin of the Alps and major mountain passes than in the
inner Alpine valleys (around 3700 m a.s.l.) that are well shielded from precipitation. On top of this
variability comes the variability due to a different aspect (Fig. 7b): On average, glaciers that are
exposed to the south have median elevations that are about 400 m higher (at 3200 m a.s.l.) than
north-facing glaciers (at 2800 m a.s.l.). However, the scatter is high and for each aspect the eleva-
tion variability is about 1500 m.
*Figure 7*

The graph in Fig. 8 shows the hypsometry of glacier area in the four countries and for the total ar-
ea in relative terms. On average, the highest area share is found around the mean elevation of 3000
m a.s.l. By referring for each country to the total area as 100%, differences among them can be
seen. Most notable is the smaller elevation range and larger peak of glaciers in Austria, the broader
vertical distribution in Switzerland (with the lowest peak value), and the slightly higher peak of the
distribution in Italy (at 3100 m a.s.l). The hypsometry of glaciers in France is closest to the curve
for the entire Alps.
*Figure 8*

**5.2 Area changes**
For a selection of 2873 glaciers present in both inventories, total glacier area shrunk from 2060
km$^2$ in 2003 to 1783 km$^2$ in 2015/16 or by -13.2% (-1.1%/a). Considering the assumed missing
area in the 2003 inventory of about 40 km$^2$ (glaciers with area gain are 29.4 km$^2$ larger in 2015/16
than in 2003), a more realistic area loss is -15% or -1.3%/a. This is about the same pace as report-



ed earlier by Paul et al. (2004) for the Swiss Alps from 1985 to 1998/99 (-1.4%/a). An example of
the strong glacier shrinkage in Austria is depicted in Fig. 9. Closer inspection of this image also
reveals the small shift (to the SE) of the S2 scenes compared to the earlier Landsat TM scenes.
*Figure 9*
The comparison of glacier outlines in Fig. 10 illustrate for the region around Sonnblickkees in
Austria why we do not provide a scatterplot of relative area changes *vs.* glacier size or country
specific area change values (cf. also Fig. 4d for Gavirolas Glacier in Switzerland). Due to the dif-
ferent interpretations in the new inventory, 125 mostly very small glaciers are 100% to 630% larg-
er than in 2003 and a large number (557) is 0% to 100% larger. For example, the 4 km$^2$ Suldenfer-
ner has increased in size by 550% as a small tributary (that holds the ID for the glacier) was dis-
connected in 2003 but is now connected to the entire glacier. Although such cases can be manually
adjusted, it would not solve the general problem of the different interpretation. For example, the
glacier in Fig. 4d has increased its size from 2003 to 2015 by 56% due to the new interpretation.
On the other hand, Careser glacier, which fragmented in six ice bodies from 2003 to 2015, lost
55% of its area when summing up all parts as opposed to 63% when considering the largest glacier
only. In consequence, the possible area reduction due to melting is partly compensated by the more
generous interpretation of glacier extents and thus with a limited meaning on the basis of individu-
al glaciers. Assuming that some glaciers in 2015/16 are larger due to included seasonal snow, the
real area loss would be even higher than the previously estimated -15%.
*Figure 10*
**5.3 Uncertainties**
**5.3.1 Glacier outlines**
The multiple digitising experiment revealed several interesting albeit well-known results. Overall,
the area uncertainty (one standard deviation, STD) is 3.3% across all participants for the total of
the digitised area (Table 3). As two glaciers (11 and 13) were not mapped by one participant, the
missing values are replaced with the mean value from the other participants. Across all glaciers but
for individual participants the uncertainty (comparing the values from the four digitisation rounds)
is considerably lower (1% to 2.7%), indicating that the digitising is more consistent when per-
formed by the same person. The area values of participant 1 (P1) are systematically higher than for
the other participants, about 6% for the total area. A detailed analysis of the digitised outlines (Fig.
11) revealed that the differences are mostly due to the more generous inclusion of debris-covered
glacier ice for two of the larger glaciers (Nr. 1 and 5). When excluding P1, the STD across the oth-
er participants is three times smaller (1.1%). The uncertainty also slightly depends on glacier size,
showing values between 1% and 6% for glaciers larger than 1 km$^2$ and between 2% and 20% for
glacier <1 km$^2$. The smallest glacier in the sample is smaller than 0.1 km$^2$ and shows variations in
STD between 8% and 44%, in the latter case also due to a reinterpretation of its extent when using



very high-resolution imagery. For such small glaciers related changes can thus result in considera-
bly different extents.
*Table 3, Figure 11*
Moreover, for P1 and most of the other participants the digitised glacier extents increased by sev-
eral per cent after consultation of very high-resolution satellite images as available in Google Earth
and the aerial imagery from the swisstopo map server. The generally very flat and debris-covered
regions were barely visible on the S2 images and have been digitised very differently in each of
the four rounds. Hence, the possibility for a re-interpretation of the outlines within the same exper-
iment resulted in higher standard deviations. If such regions have to be included in a glacier inven-
tory or not can be discussed, as the transition to ice-cored medial or lateral moraines is often grad-
ual and including these features in a glacier inventory or not is a (personal) methodological deci-
sion. However, it points to an underestimation of glacier area also with 10 m resolution sensors
and confirms earlier recommendations to double-check all digitised glacier extents with such very
high-resolution sensors, at least for 'difficult' glaciers (e.g. Fischer et al. 2014).
The uncertainty (one STD) obtained with the buffer method is ±5% (10%) when using a 10 m (20
m) buffer. This is in line with the mean values of the uncertainties derived from the multiple digit-
ising experiment and numerous previous studies.
**5.3.2 Topographic information**
The comparison of minimum, maximum and mean glacier elevation as well as mean slope and
aspect derived from the TDX and AW3D30 DEM, revealed in particular towards smaller glaciers
larger differences. Smaller glaciers are more likely to be impacted by artefacts as these easily share
a large percentage of their total area. Differences in mean slope and aspect are generally small but
increase towards larger slope values for the former. This is in agreement with the general observa-
tions that DEM quality is reduced at steep slopes. Minimum elevation is slightly higher in the
TDX DEM, which can be explained by glacier retreat between the acquisition dates. However, a
clearly lower mean elevation due an overall surface lowering of the glaciers could not be observed,
indicating that the differences are in the uncertainty range. Apart from artefacts, the uncorrected
radar penetration of the TDX DEM might play a role here as well.
# 6. Discussion
The derived size class distribution (Fig. 5) and topographic information are typical for glaciers in
mid-latitude mountain ranges with numerous smaller glaciers surrounding a few larger ones. Only
354 out of 4394 glaciers (8%) are larger than 1 km$^2$ and nearly one half (46%) is smaller than 0.05
km$^2$ covering 2.7% of the area. It might be well possible that many of the latter are no longer glac-





iers but just perennial snow and firn patches. However, for consistency with earlier inventories
they have been included. Mean elevation values do not really depend on glacier size for such glac-
iers, indicating that they can survive at different elevations and precipitation amounts have a lim-
ited impact. If they are well protected from solar radiation (e.g. by shadow or debris cover) such
glaciers might persist for some time despite increasing air temperatures. Glacier mean elevation
(about 3000 m a.s.l.) does not depend on glacier size but is modified by glacier location with re-
spect to precipitation sources and mean aspect, in particular for larger glaciers (Fig. 7).

Widespread glacier thinning and steep terrain resulted lately in interrupted profiles for several
larger valley glaciers whose lower parts are no longer nourished by ice from above. In other
words, these parts are not regenerated glaciers but melt away as dead ice. Strictly speaking, such
lower dead ice bodies (that can persist due to debris cover for a very long time) should be excluded
from a glacier inventory (Raup and Khalsa 2007). However, for consistency with former invento-
ries and their contribution to run-off we included them here and merged their IDs to obtain more
reasonable topographic information for the combined extent. Calculating this instead for the indi-
vidual parts would result in related outliers and a more difficult analysis of trends. At best, such
separated parts are identified with a flag in the attribute table, for example as a further extension to
the 'Form' attribute (e.g. '4: Separated glacier part') used in the RGI (RGI consortium 2017).
However, the differentiation from a regenerated glacier might sometimes be difficult.

Due to the differences in interpretation (Fig. 10) we have not compared the 2003 extents of indi-
vidual glaciers directly with those from the new inventory but only the total area of glaciers ob-
served in both inventories. Considering the underestimated glacier area in 2003 (e.g. due to miss-
ing debris cover) and possibly overestimated sizes in 2015 (e.g. due to included snow) the pace of
shrinkage (-1.3% /a) has not changed compared to the earlier mid-1980s to 2003 period. This indi-
cates that most glaciers have not yet reached a geometry that is compliant with current climate
conditions and will thus continue shrinking in the future. This becomes also clear from the snow
cover remaining near the end of the ablation period on the glaciers, covering barely 20% to 30% of
the area (e.g. Figs. 9 and 11). Assuming a required 60% coverage of their accumulation area, glac-
iers in the Alps have to lose another 50% to 70% of their area to reach again balanced mass budg-
ets (Carturan et al. 2013). There are other regions in the world with similar high (or even higher)
area loss rates such as the tropical Andes (e.g. Rabatel et al. 2013), but to a large extent this is also
due to the smaller glaciers in this region. A realistic comparison across regions would only be pos-
sible when change rates of identical size classes are compared.

The multiple digitising experiment (Fig. 11) revealed a large variability in the interpretation of de-
bris-covered glaciers among the analysts but high consistency in the corrections where boundaries
are well visible. Related area uncertainties can be high for very small glaciers (>20%) but are gen-



erally <5%. The here derived area reduction of about -15% since 2003 is thus significant, but for
small and/or debris-covered glaciers the area uncertainty can be similar to the change, making it
less reliable. However, this strongly depends on the specific glacier characteristics and cannot be
generalized to all small glaciers.

The gradual disappearance of ice under debris cover and the separation of low-lying glacier
tongues on steep slopes are major problems for any glacier inventory created these days. We de-
cided to re-connect disconnected glacier parts by their ID (to so-called *multi-part polygons*) for
consistency with earlier inventories. However, keeping them separated is another possibility, given
that possible dead ice is clearly marked in the attribute table.

# 7. Conclusions
We presented the results of a new glacier inventory for the entire Alps derived from Sentinel-2
images of 2015 and 2016. In total, 4394 glaciers >0.01 km$^2$ covering an area of 1806 ±60 km$^2$ are
mapped. This is a reduction of about 300 km$^2$ or -15% (-1.3%/a) compared to the previous Alpine-
wide inventory from 2003. The pace of glacier shrinkage in the Alps remained about the same
since the mid-1980's, indicating that glaciers will continue to shrink under current climatic condi-
tions. Due to the differences in interpretation, we have not performed a glacier-by-glacier compari-
son of area changes. The on-going glacier decline also results in increasingly difficult glacier iden-
tification (under debris cover) and topologic challenges for a database (when glaciers split). The
former is confirmed by the results of the uncertainty assessment, showing a large variability in the
interpretation of glacier extents when conditions are challenging. Despite the additional workload,
we think this is the best way to provide an uncertainty value for such a highly corrected and
merged dataset. In any case, the outlines from the new inventory should be more accurate than for
2003, as we here used the previous, high-quality national inventories as a guide for interpretation,
performed corrections by the respective experts, and worked with the higher resolution of Senti-
nel-2 data that helped in identifying important spatial details.

The clean-ice mapping with the band ratio method is straightforward, but requires well-thought
decisions on the two thresholds as they will always be a compromise. They should be tested in re-
gions with ice in cast shadow and selected in a way that the workload for manual corrections is
minimised. If a precise DEM is available, the required corrections of wrongly mapped ice in shad-
ow can be reduced as the further pre-processing for glaciers in Austria revealed. However, reduced
DEM quality and co-registration issues as well as local illumination differences can limit the bene-
fits of a topographic normalisation of the images. Due to the artefacts in the first version of the
TanDEM-X DEM, we used the ALOS AW3D30 DEM to derive topographic information for each
glacier despite the less good temporal agreement. To conclude, we had much better datasets avail-
able for this inventory compared to the 2003 dataset, but for several reasons (e.g. debris cover,



clouds, seasonal snow) the creation of glacier inventories from satellite data and a DEM remains a
challenging task with high workload and expert knowledge required.

## 8. Data availability

The dataset can be downloaded from: https://doi.pangaea.de/10.1594/PANGAEA.909133 (Paul et
al., 2019).

**Author contributions**

FP designed the study, prepared the raw glacier outlines, performed various calculations and wrote
the manuscript. PR performed most of the GIS-based calculations and the editing that was required
to obtain a complete dataset and change assessment (e.g. drainage divides, satellite footprints,
country boundaries, DEM mosaicking and co-registration, dataset merging, topographic data). All
authors processed, corrected and checked the created glacier outlines in their country and contrib-
uted to the contents and editing of the manuscript. FP, DF, JN, AR, and PR performed the multiple
digitising of glacier outlines for uncertainty assessment.

**Competing interest**

The authors declare that they have no conflict of interests.

**Acknowledgements**

This study has been performed in the framework of the project Glaciers_cci (4000109873/14/I-
NB) and the Copernicus Climate Change Service (C3S) that is funded by the European Union and
implemented by ECMWF. R.S. Azzoni and D.Fugazza were funded by DARA - Department for
regional affairs and autonomies of the Italian presidency of the council of Ministers (funding code
COLL_MIN15GDIOL_M) and Levissima Sanpellegrino S.P.A., (funding code
LIB_VT17GDIOL). For the French Alps contribution, A. Rabatel and M. Ramusovic acknowledge
the *Service National d'Observation* GLACIOCLIM (Univ. Grenoble Alpes, CNRS, IRD, IPEV,
https://glacioclim.osug.fr/), the LabEx OSUG@2020 (*Investissements d'avenir* – ANR10
LABX56), the EquipEx GEOSUD (*Investissements d'avenir* – ANR-10-EQPX-20), the CNES /
Kalideos Alpes and CNES / SPOT-Image ISIS program #2011-513 for providing the Pléiades im-
ages and SPOTDEM from 2011. For the Austrian Alps, G. Schwaizer and J. Nemec acknowledge
funding from the Environmental Earth Observation (ENVEO) IT GmbH and the Austrian Re-
search Promotion Agency (FFG) within the ASAP9-SenSAP project (3574408).



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





# Tables

*Table 1: Details about the Sentinel-2 tiles used to create the inventory, C.: Country.*

| Nr. | Tile | Date | C. | Nr. | Tile | Date | C. | Nr. | Tile | Date | C. |
|---|---|---|---|---|---|---|---|---|---|---|---|
| 1 | 32TMT | 29 8 15 | CH | 9 | 32TMS | 23 8 16 | IT | 17 | 31TGK | 29 8 15 | FR |
| 2 | 32TNT | 29 8 15 | CH | 10 | 32TNS | 26 8 15 | CH, AT | 18 | 32TLR | 29 8 15 | FR, IT |
| 3 | 32TNT | 26 8 15 | AT | 11 | 32TNS | 29 9 16 | IT | 19 | 32TLR | 7 10 17 | IT |
| 4 | 32TPT | 26 8 15 | AT | 12 | 32TPS | 29 9 16 | IT | 20 | 32TMR | 7 10 17 | IT |
| 5 | 32TQT | 27 8 16 | AT, IT | 13 | 32TPT | 26 9 16 | IT | 21 | 31TGK | 29 8 15 | FR |
| 6 | 33TUN | 27 8 16 | AT, IT | 14 | 32TQS | 7 8 16 | IT | 22 | 32TLQ | 23 8 16 | IT |
| 7 | 32TLS | 29 8 15 | CH, FR | 15 | 32TQS | 27 8 16 | IT | 23 | 32TLP | 29 8 15 | IT |
| 8 | 32TMS | 2 8 15 | CH | 16 | 33TUM | 2 8 17 | IT | | | | |


*Table 2: Glacier area and count per size class for the entire sample.*

| Size class [km²] | 0.01-0.02 | 0.02-0.05 | 0.05-0.1 | 0.1-0.2 | 0.2-0.5 | 0.5-1 | 1-2 | 2-5 | 5-10 | 10-20 | >20 | All |
|---|---|---|---|---|---|---|---|---|---|---|---|---|
| Count | 966 | 1060 | 723 | 532 | 520 | 244 | 177 | 103 | 48 | 16 | 5 | 4394 |
| Count [%] | 22.0 | 24.1 | 16.5 | 12.1 | 11.8 | 5.6 | 4.0 | 2.3 | 1.1 | 0.4 | 0.1 | 100 |
| Area [km²] | 13.83 | 34.44 | 51.42 | 75.48 | 163.87 | 168.28 | 249.06 | 319.13 | 322.96 | 211.85 | 195.56 | 1805.8 |
| Area [%] | 0.8 | 1.9 | 2.8 | 4.2 | 9.1 | 9.3 | 13.8 | 17.7 | 17.9 | 11.7 | 10.8 | 100 |


*Table 3: Results of the multiple digitising experiment, listing for each of the five*
*participants the mean glacier area (in km²) in the columns P1 to P5 along with the*
*standard deviation in per cent (STD%). The last two columns provide the averaged values*
*across all participants for each glacier and the last row gives total areas and their*
*standard deviation across all glaciers and for each participant. The two values marked in*
*blue are mean values derived from the other four participants. Red values mark highest*
*values for glaciers larger and smaller than 1 km². Glacier ID 4 is missing as it was*
*digitised as one glacier (with ID 5) by most participants.*

| Gl.-ID | P1 | STD% | P2 | STD% | P3 | STD% | P4 | STD% | P5 | STD% | Mean | STD% |
|---|---|---|---|---|---|---|---|---|---|---|---|---|
| 1 | 9.37 | 1.89 | 8.96 | 0.18 | 8.40 | 0.79 | 8.77 | 0.99 | 8.64 | 3.86 | 8.83 | 4.14 |
| 2 | 6.50 | 2.10 | 6.08 | 1.31 | 6.07 | 1.43 | 5.95 | 0.81 | 6.25 | 1.31 | 6.17 | 3.48 |
| 3 | 0.79 | 3.75 | 0.72 | 3.51 | 0.65 | 1.62 | 0.73 | 0.74 | 0.71 | 8.77 | 0.72 | 7.02 |
| 5 | 4.10 | 3.03 | 3.22 | 2.33 | 3.50 | 3.92 | 3.45 | 5.66 | 3.45 | 7.46 | 3.54 | 9.33 |
| 6 | 2.88 | 1.82 | 2.83 | 1.52 | 2.90 | 3.32 | 2.75 | 2.69 | 2.91 | 1.86 | 2.85 | 2.27 |
| 7 | 1.20 | 1.04 | 1.06 | 6.10 | 1.16 | 2.71 | 1.14 | 1.91 | 1.20 | 2.90 | 1.15 | 4.81 |
| 8 | 5.35 | 0.24 | 5.13 | 1.58 | 5.25 | 0.77 | 5.24 | 0.31 | 5.26 | 1.24 | 5.25 | 1.51 |
| 9 | 2.75 | 0.43 | 2.75 | 1.64 | 2.59 | 3.80 | 2.72 | 2.17 | 2.64 | 1.53 | 2.69 | 2.64 |
| 10 | 0.38 | 6.38 | 0.30 | 2.76 | 0.25 | 4.37 | 0.30 | 3.39 | 0.25 | 4.80 | 0.30 | 17.24 |
| 11 | 0.28 | 12.40 | 0.27 | 0.64 | 0.26 | 2.06 | 0.26 | 1.71 | 0.30 | 8.69 | 0.27 | 6.77 |
| 12 | 0.24 | 1.41 | 0.25 | 4.34 | 0.20 | 3.30 | 0.21 | 5.54 | 0.23 | 6.79 | 0.23 | 8.85 |
| 13 | 0.08 | 41.67 | 0.12 | 17.80 | 0.03 | 8.00 | 0.08 | 17.68 | 0.11 | 17.65 | 0.08 | 44.21 |
| 14 | 0.21 | 4.29 | 0.17 | 15.52 | 0.11 | 16.16 | 0.20 | 5.03 | 0.21 | 13.42 | 0.18 | 24.01 |
| 15 | 0.12 | 4.96 | 0.12 | 7.10 | 0.11 | 1.09 | 0.11 | 14.22 | 0.14 | 3.45 | 0.12 | 11.01 |
| Sum | 34.25 | 1.48 | 31.97 | 0.97 | 31.48 | 1.13 | 31.90 | 0.91 | 32.31 | 2.72 | 32.38 | 3.35 |





## Figure captions

Fig. 1: Overview of the study region with footprints (colour-coded for acquisition year) of the Sen-
tinel-2 tiles used (see Table 1 for numbers).

Fig. 2: Comparison of hillshade views from a) the AW3D30 DEM and b) the TanDEM-X DEM
for a region around the Mt. Blanc/Monte Bianco. Glacier outlines are shown in red, data voids in
the TanDEM-X DEM are depicted as constantly grey areas. The AW3D30 DEM has been ob-
tained from https://www.eorc.jaxa.jp/ALOS/en/aw3d30/index.htm and is provided by JAXA. The
TanDEM-X DEM has been acquired by the TerraSAR-X/TanDEM-X mission and is provided by
DLR (DEM_GLAC1823).

Fig. 3: Results of the automated (clean ice) glacier mapping and threshold selection. a) band ratio
MSI band 4 / MSI band 11 (red/SWIR). b) Glacier classification results using different thresholds.
The lower values add some additional pixels, in particular in shadow regions where the threshold
is most sensitive. c) Blue band threshold to remove wrongly classified rock in shadow. The highest
value has been used resulting in a good performance in the left part of the image (white arrow) and
a bad one to the right (green arrow), where correctly classified ice in shadow is removed. d) Final
outlines (light blue) on top of the Sentinel-2 image in natural colours. All Sentinel-2 images shown
in the background: © Copernicus data (2016).

Fig. 4: Examples of challenging classifications in different countries. a) Debris cover delineation
(red) around Grossvenediger (Hohe Tauern) in Austria with raw extents (light grey) and outlines
from the previous national inventory (yellow). b) Tré-La-Tête Glacier (Mont-Blanc) with automat-
ically derived glacier extents (green), manually corrected outlines from 2015 (red) and outlines
derived from aerial photographs taken in 2008 (yellow). The S2 image from August 2015 is in the
background. c) Subset of the Orobie Alps in Italy (S2 image from September 2016), with evidence
of topographic shadow and debris covered glaciers. The inset shows an aerial photograph with bet-
ter glacier visibility but seasonal snow. d) S2 image from 2015 showing differences in interpreta-
tion of debris cover for Gavirolas glacier in Switzerland for the inventories from 2003 (yellow),
2008 (green) and 2015 (red). The inset shows a close-up of its lowest debris-covered part obtained
from aerial photography for comparison (this image is a screenshot from Google Earth). All Senti-
nel-2 images shown in the background: © Copernicus data (2016).

Fig. 5: Relative frequency histograms for glacier count and area per a) size class and b) aspect sec-
tor for all glaciers.

Fig. 6: Glacier area vs. a) minimum and maximun elevation and b) elevation range for all glaciers.






Fig. 7: Spatial distribution of median elevation (colour coded) for glaciers larger 0.5 km². The inset
shows a scatterplot depicting glacier aspect (counted from North at 0/360º) vs. median elevation.

Fig. 8: Normalised glacier hypsometry per country as derived from the AW3D30 DEM.

Fig. 9: Visualisation of the strong glacier area shrinkage between 2003 (yellow) and 2015 (red) for
a sub-region of the Zillertal Alps (Austria and Italy). Sentinel-2 image shown in the background: ©
Copernicus data (2016).

Fig. 10: Overlay of glacier outlines from 2003 (black) and 2016 (yellow) showing the different
interpretation of glacier extents for the region around Sonnblickkees (SBK) in Austria. Sentinel-2
image shown in the background: © Copernicus data (2016).

Fig. 11: Overlay of glacier outlines from the multiple digitising experiment by all participants.
Colours refer to the first (yellow), second (red), third (green) and fourth (white) round of digitisa-
tion. Sentinel-2 image shown in the background: © Copernicus data (2016).

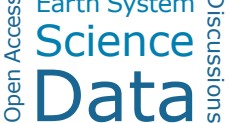

)                                              b)

# Figures

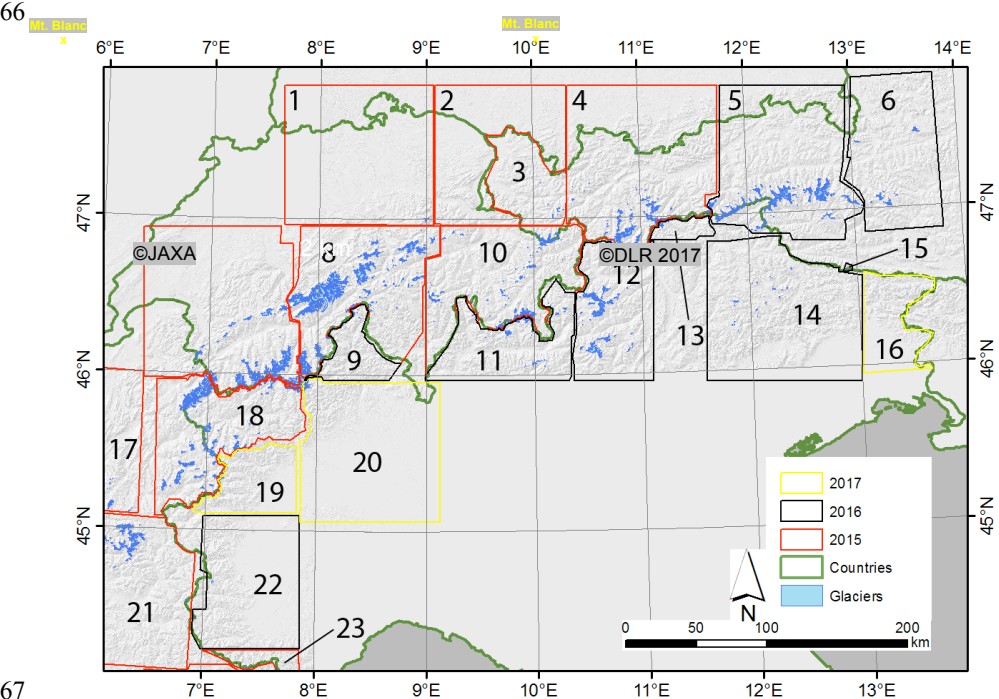

Figure 1

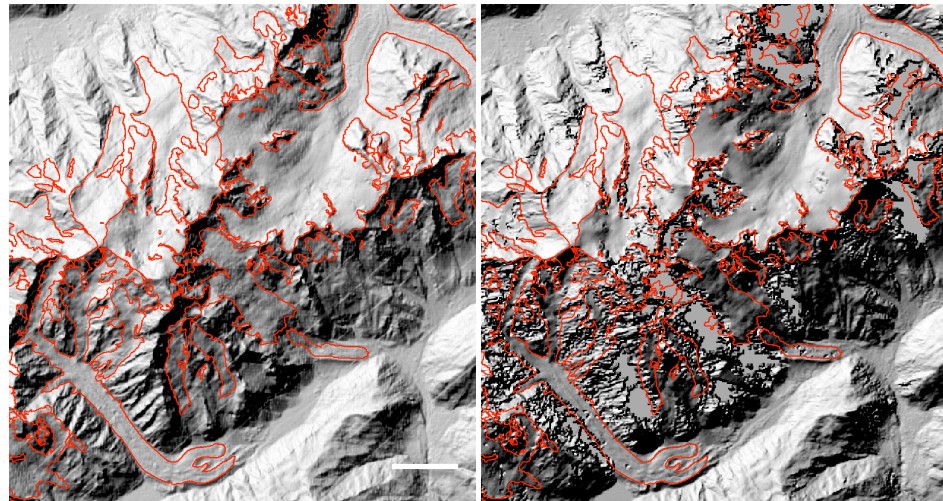

Figure 2




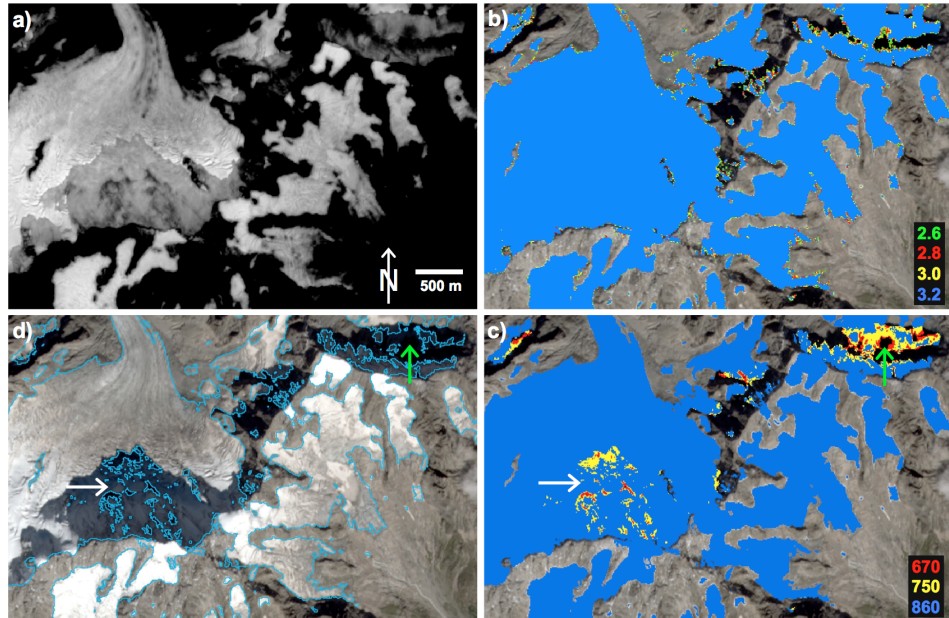


Figure 3


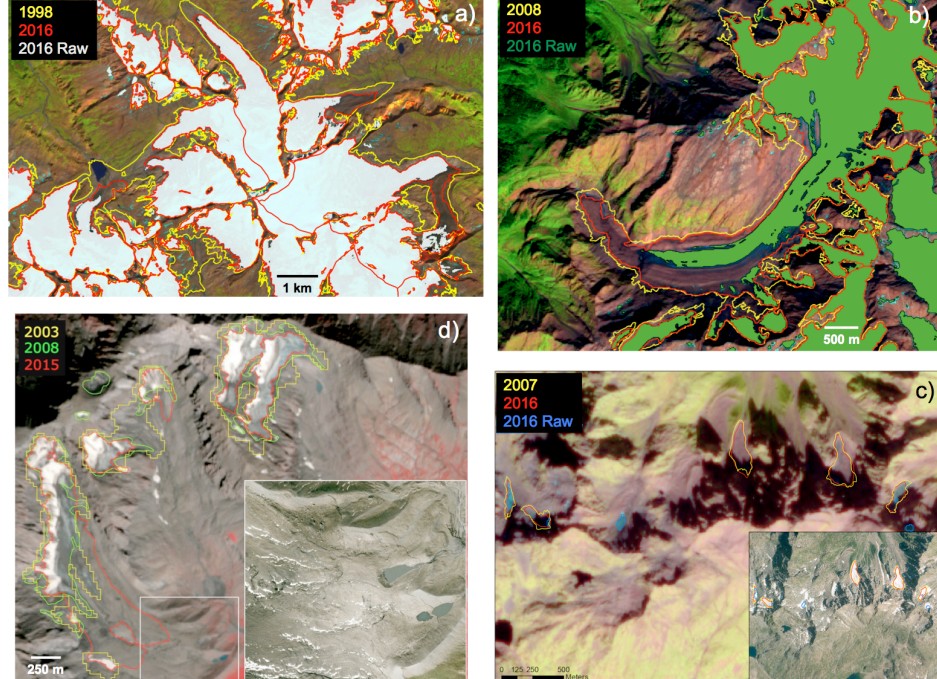


Figure 4



b)


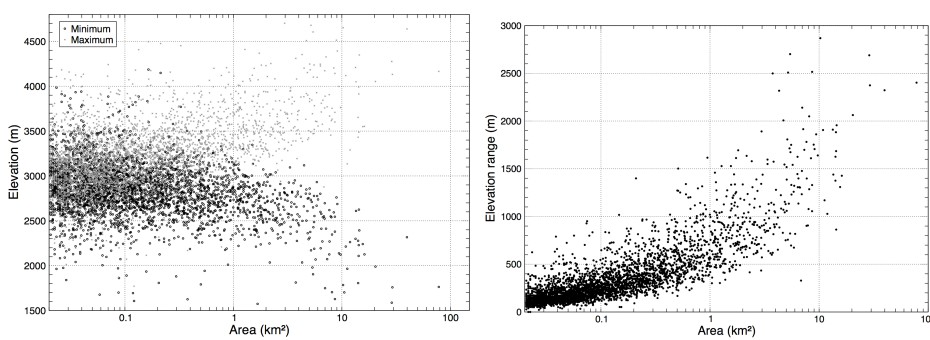


Figure 5


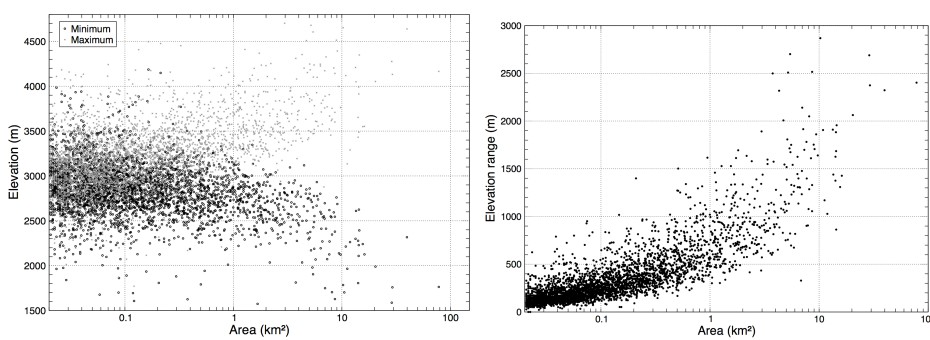


Figure 6:


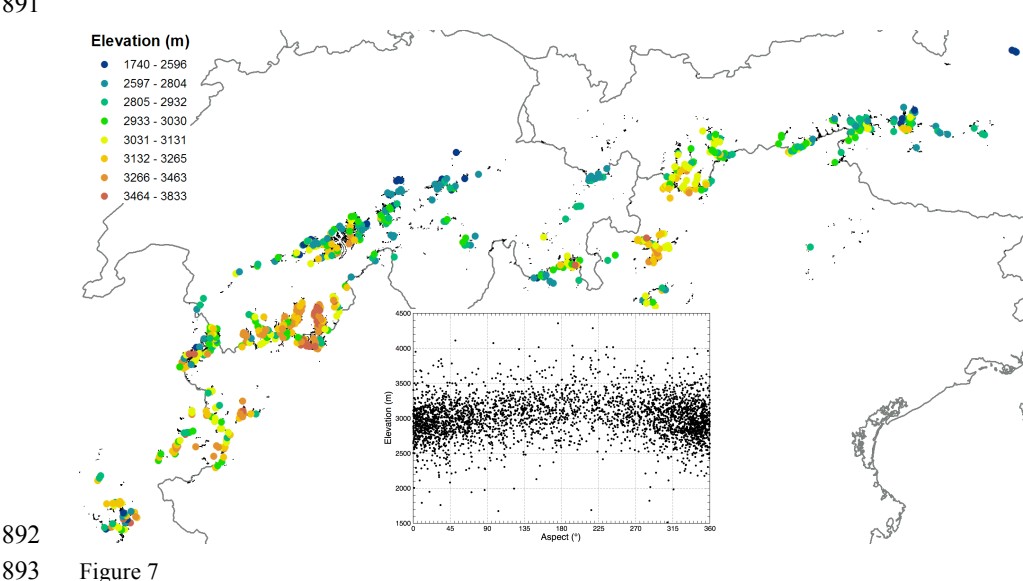


Figure 7





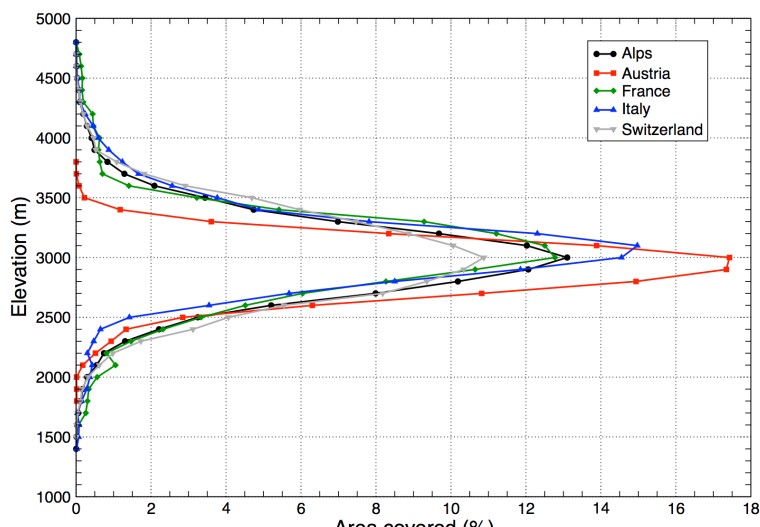


Figure 8



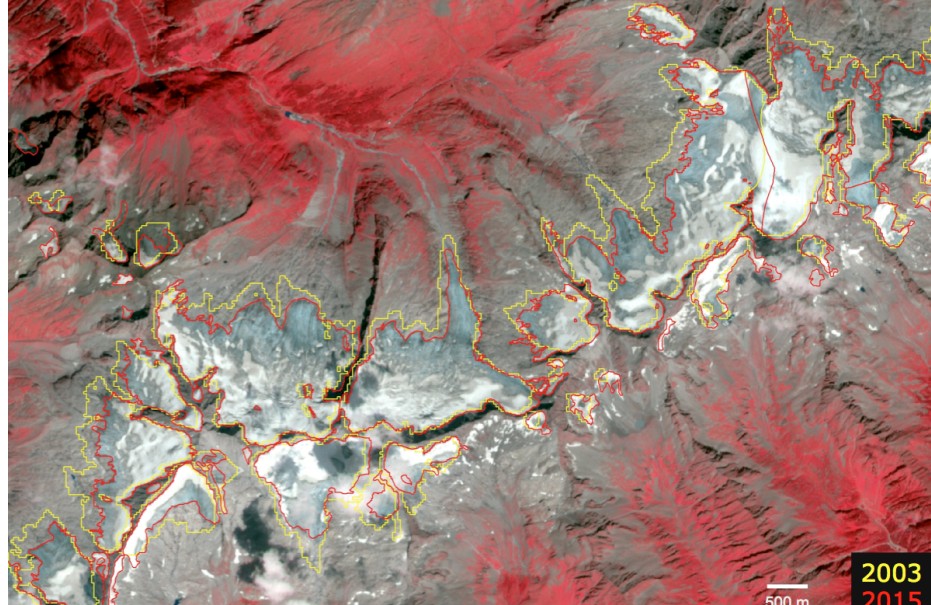


Figure 9



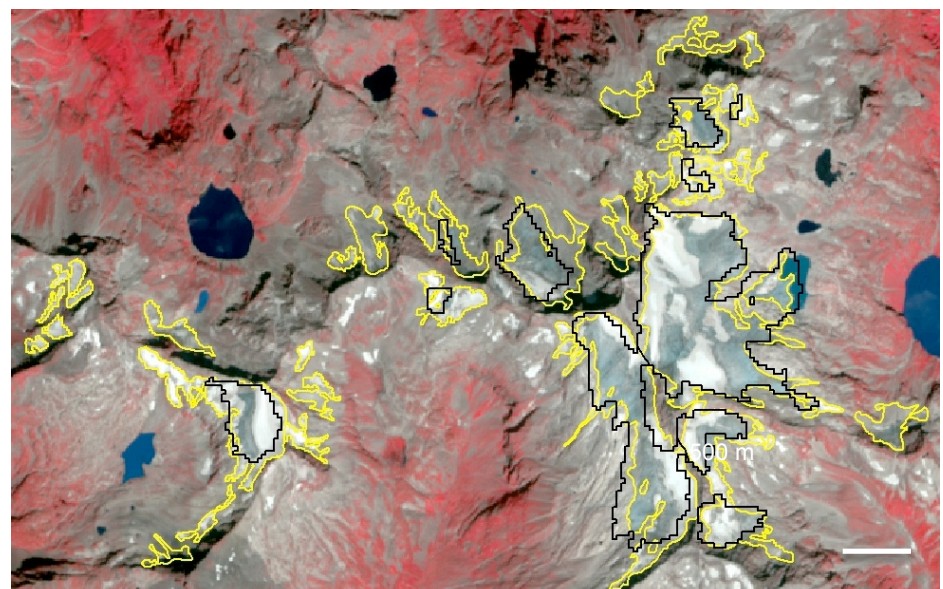

Figure 10

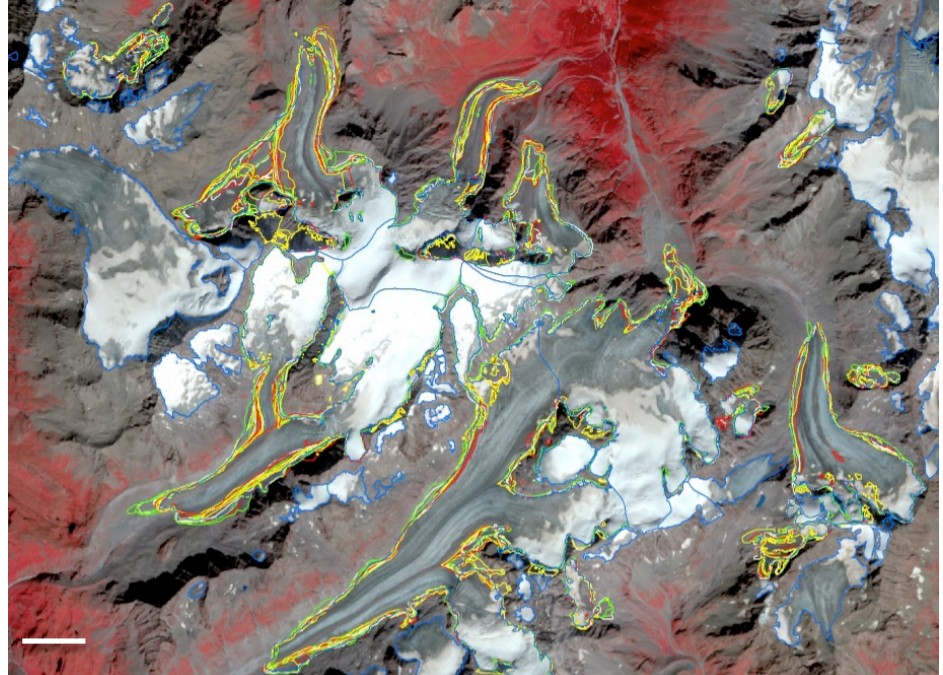

Figure 11