# Peer review of "Glacier shrinkage in the Alps continues unabated as revealed by a new glacier inventory from Sentinel-2"

_Earth System Science Data, 2019_

## Referee Comment (RC1) · Andrea Fischer (Referee) · 24 Jan 2020

The article "Glacier shrinkage in the Alps continues unabated as revealed by a new glacier inventory from Sentinel-2" presents a new compilation of glacier boundaries for the Europen Alps. The improved resolution of Sentinel data allows a more accurate delineation of glacier boundaries as the Landsat based inventory of 2003. The new inventory is based on remote sensing data acquired during 2 years only, and was based on past national inventories, corrected for inconsistencies at the national borders. Extensive error assessments ensure high data quality, even in shaded and debris covered areas. The data and methods presented are new, and the data will be used for lot of

different applications, for example hydrological modelling. Methods and materials are described in sufficient detail. Reference and citation to other data sets are correct and appropriate. A minor side remark concern the climate data cited, see details below (1) with a suggestion to add a data reference. The article is supporting the data set and contains also valuable information on accuracy and limitations. The data set is under review at Pangaea.de, which is the normal process at this platform, I am confident that the data will be available once the ESSD paper is published. I consider the examples given in the Figures sufficient to judge general data quality. I cannot check the completeness of the data, but the total area given seems to support the idea that all glaciers have been mapped. Data processing and format is state of the art. I consider the data to be of highest quality. Regarding the problem of inclusion of small glaciers and the comparison to the 2003 data, I suggest some rephrasing to better distinguish between real area changes and mapping artefacts (2), which is described somehow misleading in the current version (but understandable though knowing the problem, confusing for researcher being data users only). The data set is useful in the current format and size, with appropriate metadata. Length and structure of the article is appropriate, wording is clear. Figures and Tables have high quality and show relevant items. Finally, I understand the data set by reading the article, and will potentially use and recommend it. How well do the respective data sets presented by an article and the article itself meet the following criteria (rated 1–4, excellent–poor): Significance 1 The data is useful fulfilling the criteria of - Uniqueness: Much effort has been taken to compile a unique data set of glacier boundaries in the European Alps. Mapping methods have been improved substantially in contrast to earlier data sets. - Usefulness: The data so far available for the glacier boundaries in European Alps have been either spatially very detailed but temporally inhomogeneous or spatially coarse with an accurate time stamp. In addition to that, inconsistencies at national borders have not been corrected so far at a larger scale. The new Alpine inventory is an excellent basis for all types of glaciological, hydrological and climatological studies and will be used quite frequently.

- Completeness: The data set is the completest inventory of European Alps possible

and available. Data quality 1 As the data described in the article has been compared to our LiDAR based data set of the same area in cooperation with our staff, I can confirm the data quality although the full data set can not be downloaded right now from Pangaea. Presentation quality 1 The article is very clear and concise and describes strength as well as weaknesses of the data set.

Detailed comments: (1) line 114: The Histalp instrumental data (Auer et al., 2007) confirms the Alpine ridge as a climate trend divide in terms of precipitation. This is not necessarily a contradiction to the results of Casty et al., as the resolution and accuracy of the data set is very different. For completeness, I would recommend to cite both articles, as the HISTALP result can be used as valid hypothesis for explaining different mass balance responses North and South of the main Alpine ridge conformed by respective mass balance monitoring.

Line 416: I can confirm that small glaciers are found at all elevations, but I can not confirm that this means that they are independent of climatic parameters. Abermann et al found that the altitudinal distribution of glaciers depends on precipitation rates also, thus the distribution alone without having a look on the type of snow accumulation and radiative setting must fail. I recommend to add a deeper discussion on climate sensitivity of small glaciers or just skip that very shortened remark. This would also resolve the contradiction to lines 428 ff. (2) lines 466 and 480 I presume that the 125 very small glaciers do not really increase their size as a result of the mapping procedure as written here. I recommend to rephrase like that: The area mapped for 125 glaciers increased by XXX %, but in reality we expect that these glaciers shrinked. Also for the Suldenferner in line 470 it is not clear if the authors consider the mapping to create reliable numbers or artefacts which they want to discuss here. If the paragraph intends to warn people on working with the inventory analyzing very small samples or single glaciers, this is not entirely clear. Also the remark on the larger (compared to 2003??) glaciers 2015/16 as result of seasonal snow included leaves open if the authors consider this as an artefact or a glacier advance. In the last sentence, it is not
entirely clear what is meant by 'real' area loss: The area loss corrected for mapping artefacts? Maybe it would be easier to understand if we read here that the authors think that their estimate of area loss is rather a lower threshold considering mapping uncertainties tend to diminish the area change?

Line 561: Missing debris cover in 2003: Have the glaciers been free of debris in 2003, or were debris covered areas not mapped in 2003?

References: Abermann, J., Kuhn, M., Fischer, A., Climatic controls of glacier distribution and glacier changes in Austria. Annals of Glaciology, 52(59), 83-90, (2011).

Auer I, Böhm R, Jurkovic A, Lipa W, Orlik A, Potzmann R, Schöner W, Ungersböck M, Matulla C, Briffa K, Jones PD, Efthymiadis D, Brunetti M, Nanni T, Maugeri M, Mercalli L, Mestre O, Moisselin J-M, Begert M, Müller-Westermeier G, Kveton V, Bochnicek O, Stastny P, Lapin M, Szalai S, Szentimrey T, Cegnar T, Dolinar M, Gajic-Capka M, Zaninovic K, Majstorovic Z, Nieplova E, 2007. HISTALP – Historical instrumental climatological surface time series of the greater Alpine region 1760-2003. International Journal of Climatology 27: 17-46

---

## Referee Comment (RC2) · Sam Herreid (Referee) · 19 Feb 2020

The manuscript "Glacier shrinkage in the Alps continues unabated as revealed by a new glacier inventory from Sentinel-2" by Paul et al. presents a new glacier inventory that optimizes several criteria including consistency, a tight time-span of source data acquisition, precision, and complete coverage over the Alps. The authors provide a thorough and well written manuscript that I found easy and enjoyable to read. One unique element of this work is the lead author also authored an earlier study that derived a similar product from Landsat imagery acquired in 2003. I was surprised that, even with this degree of consistency, this new inventory was only marginally compati-

ble for a comparison. This seems largely due to the inclusion of "new" glacierized area that was not previously mapped, yet the authors walk this back some in the discussion admitting that some "glaciers" might really be perennial snow and firn patches. Since other scientists will likely base their work on this inventory, I think it is most fitting that the expert authors here make decisions and interpretations that they are confident with. Below are mostly minor comments with some concerns regarding analysis subdivided by political boundaries, results that reflect the highly variable 'number of glaciers' and the buffer method error analysis.

Abstract

L23-24 "...national inventories have been used as a guide to compile a consistent update." How do several inventories generated from different countries by different analysists enable consistency?

L23-27 It is odd that you develop the guiding datasets first and then explain methods to map glaciers from raw data. It's not perfectly clear what fraction of the new inventory is in fact new.

L24-25 You shouldn't have sentences beginning with "However" and "Whereas" back to back.

L28-29 Is there a topographic result you can add to the abstract? This is the only sentence in the abstract referencing topographic information, and it doesn't explain why this is relevant to the study.

L57 Following the logic from L47-48 shouldn't this be $\pm 3\%$?

L59 Change "has" to "was"

L63 Change "representing" to "represents"

L72 "...in part to be compliant with the analysis in earlier inventors." Do you mean communicative? What are the other parts?

L90 What does "commissioning phase" mean?

L94 "...and distribute the raw outlines to the national experts for edition of wrongly classified regions" This sounds smart to use local knowledge, but to me it does not fit the description "consistent". To me a consistent method would be done by one person or one automated routine.

L96-97 "As a guide for the interpretation the analysts used the latest high-resolution inventory in each country" What does this mean exactly? Is the time stamp then some mix of 2015 and the range spanning the imagery used for the national inventories?

Study Region

L121 "With a total area of about 2000 kmˆ2 in 2003" You say 2100 kmˆ2 on L30 in the Abstract.

L121-122 Please add a citation for "about 1 m w.e. . . . 2 Gt of ice per year."

L126 "and the mean elevation is around 3000 m a.s.l., a unique value compared to other regions of the RGI." In which direction?

L130 "...many glaciers – large and small – become invisible under increasing amounts of debris" "Invisible" is imprecise language, "indistinguishable from optical data" is better.

L132 "...mapping their extent is increasingly challenging" Is this the same glaciers becoming more challenging or new glaciers becoming challenging?

Datasets

L149-150 "only the required bands, no longer possible" I don't understand this.

L161-164 I really like this level of detail you provide. Here you make clearer what "commissioning phase" means (question to L90).

L193 Change "using" to "to use"

L194-195 "...due to the locally poor geolocation of the S2 scenes . . . the location of the ice divides was [change to: were] partly manually adjusted" Are you saying you moved correctly geolocated flow divided to agree with an erroneously geolocated S2 image? Please be clear what this implies about your new dataset (including the magnitude of these adjustments) and how you motivated your choices.

Methods

L201-207 This is confusing to follow.

L208 "We followed the recommendation to select the threshold in a way that good mapping results in regions with shadow are achieved." Where is this recommendation? Can you please add the threshold values used to Table 1 and cite that here?

L213 You mention several times "misclassified rock in shadow" which I believe means the dark rock becomes darker but is then classified as glacier, I'm confused by this.

L222 "contrast stretched" Is this just referring to how you assigned display colors? Probably unneeded information if so.

L235 "All pre-processed scenes were provided in their original geometry for correction" This doesn't make sense to me. What is the geometry of a pre-processed satellite image?

L238 Dark bare ice (that you reference in the sentence before) is also an omission error

L239 And shadowed rock classified as glacier is also a commission error. Meaning, you're not wrong but it's slightly odd to not give these aforementioned quantities the same classification.

L240 I know this is a glacier definition interpretation question, but I really cannot imagine a 0.01 kmˆ2 "glacier" internally deforming. The argument for including these patches is weak, in my opinion.

L246 "(see Paul et al. 2016)" do you mean "following Paul et al."?

L253 Regarding sub meter resolution data to inform debris cover delineation, I agree this can be very helpful, but I think it can also be less helpful. In some instances it is simply not clear, even if you were standing in the field, in these cases I think the high resolution images give us analysts a false sense of confidence. I would even argue that in some cases a lower 10-15 m resolution helps flow features stand out which is probably a better guide to finding the glacier margin than images that can resolve individual clasts. You make a similar counter argument to mine later at L321-323.

L254 "we illustrate the strong glacier shrinkage from 1998" Strong relative to what? And I believe all citations so far have been to the 2003 former outlines, please add a citation for these 1998 outlines.

L265 "changes...are important." Hard to constrain importance, I would say "notable" or "visible"

L270-284 The relevance of these two paragraphs is not clear to me. What information does your reader need from this to understand your study? I think it can be said much more concisely.

L289 Can you please let us know how common these shadow errors were? In kmˆ2 preferably but also qualitative terms could be okay. I think it is useful information for others who will use this work as a guide for their own glacier inventories.

L293 Do you mean "[Italian] alps"? also I don't think "i.e." is correct if you list the full set of 3.

L295 If you go into sub-region detail it would be helpful to have these regions labeled in Figure 1 and possibly summarized in a table.

L300-304 I think this is an appropriate place to mention rock glaciers and either cite others regarding their definition/interpretation or use these data and possibly change since 2003 to make your own inference on this classification. Were you sufficiently

convinced what you show in Figure 4c is in fact glacier? You defer to the previous inventory, but you clearly demonstrate a level of expertise in this subject and I think the readers will appreciate a more decisive call on these small, difficult glaciers.

L317 I think something like "completely ablated" is more precise than "disappeared" but maybe only stylistic.

L325-326 "This glacier has thus strongly grown since 2003 due to a new interpretation..." This is incorrect language, change to "the interpreted glacier area strongly grew"

L336-339 I agree with your statement that political boundaries are meaningless in a scientific context, yet you go on to present your results per country. I understand that the source of some degree of your results are from national inventories, but my sense was this article is a pure research stitching together of these datasets. What is your motivation to partition your results by country?

L341 I don't think "digital" is needed.

L360-362 Please provide the specific rational as to why a glacier specific comparison was not possible between glaciers that met the "point in polygon" check. One thing that makes this article so unique is that the 2003 inventory was made by you. This is the interesting point of unique consistency that you bring with this study but here it seems like you deflect from this. Has your personal definition of a glacier changed so much since the earlier inventory? There is a strange human element to this line of work/this study, which you later spend considerable effort attempting to constrain, yet this change in interpretation/definition of a glacier between the 2003 inventory and this one, with the same lead author, is surprising and interesting to me.

L382 "...we applied the buffer method..." Please add a citation for this

L390 I don't understand how it is computationally expensive to buffer the debris-covered areas (a smaller area than total glacier area) while it is computationally feasible
to buffer all areas. I have applied buffers to all glaciers and debris cover on Earth on a typical laptop without outstanding computational demand. Is there a step or condition that causes the high computational cost?

L391-392 I think Figure 4d is a good example of where this assumed realistic uncertainty estimate of $\pm 2$ pixels is not applicable. You say this depends on the degree of debris coverage which I agree with, could you apply a different buffer as a function of debris-covered area? Maybe it could help keep your above referenced computational cost low if only considering the very debris-covered glaciers.

L396-397 Please watch your significant figures, the sum your report is slightly off. Further, do you have confidence in your results in Austria to 0.01 kmˆ2?

L399-402 I do not see the value in 'number of glaciers' based results. I am not convinced a different research group repeating your study will fine the same number of glaciers. If you disagree with this argument, I would at least suggest a minimum area of 1 kmˆ2 to promote some degree of repeatability.

L415 "glaciers smaller than 1 kmˆ2 can be found at all elevations" I see what you are trying to say but this statement is incorrect.

L415-416 "indicating that their mean elevation [remove: does] only slightly depend [depends] on climate factors" This doesn't make sense to me. I think you are trying to obliquely address the question: why aren't some of these small glaciers bigger? I think there might be a statement you can make here but as it is I don't think there is supporting evidence.

L17 "arrange around a climatically driven mean elevation which is around 3000 m a.s.l." Why not compute the mean and add a fit mean line to Figure 6a? "...the largest glaciers are not those with the highest elevation range..." Yes, I actually think they are and your next statement contradict your previous statement "...and for the majority of glaciers the elevation range increases with glacier size" One very steep, small glacier

is an outlier, not relation breaking evidence.

L422 "This is typical for regions dominated by mountain and valley glaciers as these follow the given topography" This statement needs a citation and an example of regions where glaciers (or ice sheets) do not follow the given topography.

L423-424 "an exception from the rule" What rule? I would call it an outlier

L428 Can you please motivate your use of median here versus mean at L415.

L428 "largely driven by temperature, precipitation and radiation" Since you don't specify glacier size this is in contradiction to L415-416.

L428 Wouldn't topography be an important factor here?

L429 "temperature is rather similar at the same elevation over large regions" Needs citation

L431 Remove "amounts"

L434-435 Why are glaciers larger than 0.5 kmˆ2 les impacted by local topographic conditions?

L435 "...median elevations (around 2400 m a.s.l.)" L418 says 3000 m a.s.l.

L438 "b" not labeled in Figure 7

L439 "On average, glaciers...have median elevations that are about 400 m higher" Is this a mean of medians? I'm a little lost how you choose distribution middle metrics.

L440 "However, the scatter is high" It's hard to understand a "result" that is then walked back some qualitative amount. Can you draw on statistical tests to qualitatively inform signal form noise?

L445: I am again missing the scientific argument for presenting result per country.

L456 "For a selection of 2873 glaciers present in both inventories..." According to this,

the number of glaciers in 2003 was 65% of the number of glaciers now. I find this more concerning than a result. I am inclined to assume this new inventory is overly generous with a "glacier" classification and I would strongly encourage the authors to double check their confidence on what they are calling glaciers.

L460-461 Per country results should probably be in a table if you report some here.

L462 "Reveals [should be: a] small shift" Can you please say more about this translation error, it doesn't look linear(?) is it elsewhere? Should this be corrected?

L466-468 As stated above I'm concerned there is not a way to make a per glacier area change plot. In my opinion, the new outlines (yellow) in Figure 10 have a lot of unrealistic area: too narrow, too small, not following a flow pattern. For the glaciers that do intersect 2003 it is clear the outlines are of very high quality, but I think the 2003 interpretation of what is a glacier is in some cases better.

L48 "with a limited meaning on the basis of individual glaciers" These differences that disable a per glacier change analysis aggregate to the whole, I'm not sure how you can consider changes of the whole Alps or per country and not be able to consider individual glaciers.

L480 "-15%" This value starts at -13.2% earlier in the manuscript, was bumped up to 15% at L459 and now is "even higher". For the results section I think it's best to present a consistent and confident value.

L493 "a detailed analysis" Where is this analysis? Or do you mean the overlay Figure only?

L495 "When excluding P1" what if P1 is the most trained expert?

L505 "digitized glacier extents increased by several per cent after consultation of very high-resolution satellite images" Is there a way this information could be added to Table 3?

L510-513 "If such regions have to be included. . .can be discussed. . .including these features in a glacier inventory or not is a (personal) methodological decision" This belongs in the discussion section.

L522-524 Check English here.

L528 re-state acquisition gap dates.

L530 5 years of elevation change are within the uncertainty range? I would expect there to be a clear signal.

Discussion

L535 missing citation here

L536-537 Probably belongs in results section

L538-539 "However, for consistency with earlier inventories they have been included" Figure 10 makes it clear this is not true, consistency with earlier inventories would exclude non-glacier area. I think it's concerning that the authors walk back what is considered a glacier here in the discussion, in my opinion, we need this set of experts to make these (hard) decisions so the rest of the community can uses this product with confidence. L540 "precipitation amounts have a limited impact" I think this a contradiction to L431.

L543-544 I don't think Figure 7 supports these claims.

L546 "Widespread glacier thinning" L530 suggests no thinning

L546-547 confusing wording

L548 Confusing English in this sentence

L551-552 "merged their IDs. . .combined extent" This is unclear

L558-559 As stated above, interpretation/definition errors don't disappear at a wider scale.

L565 Snow covering 20% to 30%, where did these numbers come from, I don't recall anything in the methods.

L571 "change rates of identical size classes are compared" Do you mean hypsometry?

L573 "revealed a large variability in the interpretation of debris-coved glaciers" Did it? I'm not sure if this was quantified.

Conclusions

L609 "DEM quality and co-registration" were these mentioned in the text?

Tables and figures

Table 3

Am I reading correctly that STD derived for n=4? Is glacier ID 4 two glaciers in the inventory? This sounds like an interesting result that most participants called in one. Is this a unique case, or common?

Figure 3

The blue is a little misleading, possibly better as 4 sets of colored lines It's a little hard to tell what is error and what is true.

Figure 4

Are blue in a) and green in b) the same? Why different colors?

Figure 9 and 10

Many of the small glaciers identified in these figures do not look like glaciers to me.
* * *

---

## Author Comment (AC1) · 6 Apr 2020

**Response to the comments by Andrea Fischer**

The article "Glacier shrinkage in the Alps continues unabated as revealed by a new glacier inventory from Sentinel-2" presents a new compilation of glacier boundaries for the European Alps. The improved resolution of Sentinel data allows a more accurate delineation of glacier boundaries as the Landsat based inventory of 2003. The new inventory is based on remote sensing data acquired during 2 years only, and was based on past national inventories, corrected for inconsistencies at the national borders. Extensive error assessments ensure high data quality, even in shaded and debris covered areas. The data and methods presented are new, and the data will be used for lot of different applications, for example hydrological modelling. Methods and materials are described in sufficient detail. Reference and citation to other data sets are correct and appropriate. A minor side remark concern the climate data cited, see details below (1) with a suggestion to add a data reference. The article is supporting the data set and contains also valuable information on accuracy and limitations. The data set is under review at Pangaea.de, which is the normal process at this platform, I am confident that the data will be available once the ESSD paper is published. I consider the examples given in the Figures sufficient to judge general data quality. I cannot check the completeness of the data, but the total area given seems to support the idea that all glaciers have been mapped. Data processing and format is state of the art. I consider the data to be of highest quality. Regarding the problem of inclusion of small glaciers and the comparison to the 2003 data, I suggest some rephrasing to better distinguish between real area changes and mapping artefacts (2), which is described somehow misleading in the current version (but understandable though knowing the problem, confusing for researcher being data users only). The data set is useful in the current format and size, with appropriate metadata. Length and structure of the article is appropriate, wording is clear. Figures and Tables have high quality and show relevant items. Finally, I understand the data set by reading the article, and will potentially use and recommend it.

*Thank you very much for this assessment!*

How well do the respective data sets presented by an article and the article itself meet the following criteria (rated 1–4, excellent–poor): Significance 1 The data is useful fulfilling the criteria of - Uniqueness: Much effort has been taken to compile a unique data set of glacier boundaries in the European Alps. Mapping methods have been improved substantially in contrast to earlier data sets. - Usefulness: The data so far available for the glacier boundaries in European Alps have been either spatially very detailed but temporally inhomogeneous or spatially coarse with an accurate time stamp. In addition to that, inconsistencies at national borders have not been corrected so far at a larger scale. The new Alpine inventory is an excellent basis for all types of glaciological, hydrological and climatological studies and will be used quite frequently.

*Thank you, we hope it will serve its purpose.*

- Completeness: The data set is the completest inventory of European Alps possible and available. Data quality 1 As the data described in the article has been compared to our LiDAR based data set of the same area in cooperation with our staff, I can confirm the data quality although the full data set can not be downloaded right now from Pangaea. Presentation quality 1 The article is very clear and concise and describes strength as well as weaknesses of the data set.

*Thank you!*

**Detailed comments:**

(1) line 114: The Histalp instrumental data (Auer et al., 2007) confirms the Alpine ridge as a climate trend divide in terms of precipitation. This is not necessarily a contradiction to the results of Casty et al., as the resolution and accuracy of the data set is very different. For completeness, I would recommend to cite both articles, as the HISTALP result can be used as valid hypothesis for explaining different mass balance responses North and South of the main Alpine ridge conformed by respective mass balance monitoring.

*We fully agree and have added the suggested reference.*

Line 416: I can confirm that small glaciers are found at all elevations, but I can not confirm that this means that they are independent of climatic parameters. Abermann et al. found that the altitudinal distribution of glaciers depends on precipitation rates also, thus the distribution alone without having a look on the type of snow accumulation and radiative setting must fail. I recommend to add a deeper discussion on climate sensitivity of small glaciers or just skip that very shortened remark. This would also resolve the contradiction to lines 428 ff.

*We fully agree that writing 'they are independent of climatic parameters' is incorrect and have actually written that their mean elevation 'does only slightly depend on climatic factors'. However, this might still be perceived incorrect so we have now written: "are also impacted by factors other than climate". We have also adjusted L428, writing now 'of larger glaciers'' instead 'of a glacier'.*

(2) lines 466 and 480 I presume that the 125 very small glaciers do not really increase their size as a result of the mapping procedure as written here. I recommend to rephrase like that: The area mapped for 125 glaciers increased by XXX %, but in reality we expect that these glaciers shrinked.

*Yes, the growth is (very likely) only due to the different interpretation (that is partly due to the different visibility of details in the 10 m Sentinel-2 images). Unfortunately, we cannot confirm if they have decreased their size or not as this has not been determined. We think so, yes, but this would be rather speculative as some small glaciers show a very limited shrinkage (see point before).*

Also for the Suldenferner in line 470 it is not clear if the authors consider the mapping to create reliable numbers or artefacts which they want to discuss here. If the paragraph intends to warn people on working with the inventory analyzing very small samples or single glaciers, this is not entirely clear.

*This is more a note that very small differences in interpretation can lead to a very different glacier extent when topological changes (e.g. two parts merging) are not considered. Collectively both extents are very similar, but the automated assessment of area changes via ID comparison cannot be performed due to the new topology. It is thus more a warning on performing such calculations automatically rather than looking in detail at the quality of the input datasets.*

Also the remark on the larger (compared to 2003??) glaciers 2015/16 as result of seasonal snow included leaves open if the authors consider this as an artefact or a glacier advance.

*As seasonal snow outside a glacier should be excluded, this is an artefact.*

In the last sentence, it is not entirely clear what is meant by 'real' area loss: The area loss corrected for mapping artefacts? Maybe it would be easier to understand if we read here that the authors think that their estimate of area loss is rather a lower threshold considering mapping uncertainties tend to diminish the area change?

*Yes, this is a good suggestion to clarify what we mean. We have now written: "Overall, glacier extents in the 2015/16 inventory might be somewhat larger than in reality due to the*

*inclusion of seasonal/perennial snow in some regions. The -15% area loss mentioned above can thus be seen as a lower bound estimate."*

Line 561: Missing debris cover in 2003: Have the glaciers been free of debris in 2003, or were debris covered areas not mapped in 2003?
*For some very few glaciers in the 2003 inventory a part of the debris cover was not included as a later comparison with higher resolution imagery revealed. For this reason the 2003 area of the mapped glaciers was underestimated. Additionally, several very small glaciers in north-eastern Italy (Venetia/Friuli) were not mapped in 2003.*

---

## Author Comment (AC2) · 6 Apr 2020

**Response to the comments by Sam Herreid**

The manuscript "Glacier shrinkage in the Alps continues unabated as revealed by a new glacier inventory from Sentinel-2" by Paul et al. presents a new glacier inventory that optimizes several criteria including consistency, a tight time-span of source data acquisition, precision, and complete coverage over the Alps. The authors provide a thorough and well written manuscript that I found easy and enjoyable to read. One unique element of this work is the lead author also authored an earlier study that derived a similar product from Landsat imagery acquired in 2003. I was surprised that, even with this degree of consistency, this new inventory was only marginally compatible for a comparison. This seems largely due to the inclusion of "new" glacierized area that was not previously mapped, yet the authors walk this back some in the discussion admitting that some "glaciers" might really be perennial snow and firn patches. Since other scientists will likely base their work on this inventory, I think it is most fitting that the expert authors here make decisions and interpretations that they are confident with. Below are mostly minor comments with some concerns regarding analysis subdivided by political boundaries, results that reflect the highly variable 'number of glaciers' and the buffer method error analysis.

*We would like to thank Sam Herreid for the careful and constructive review!*

**Abstract**

L23-24 "...national inventories have been used as a guide to compile a consistent update." How do several inventories generated from different countries by different analysists enable consistency?

*Very good point, writing it this way is clearly misleading. It should mean 'consistent within each country' so with their earlier inventories. We have added this important point. As slightly different rules had been applied previously in each country, the new inventory is not alpine-wide consistent (with the related consequences for a comparison to 2003).*

L23-27 It is odd that you develop the guiding datasets first and then explain methods to map glaciers from raw data. It's not perfectly clear what fraction of the new inventory is in fact new.

*It is basically all new. The analysts only used the outlines of the previous inventories as a guide for the manual corrections. We have moved the 'Whereas' sentence forward to better reflect the logic of the workflow.*

L24-25 You shouldn't have sentences beginning with "However" and "Whereas" back to back.

*Agreed and changed.*

L28-29 Is there a topographic result you can add to the abstract? This is the only sentence in the abstract referencing topographic information, and it doesn't explain why this is relevant to the study.

*We have now added the median elevation and its dependence on location and aspect.*

L57 Following the logic from L47-48 shouldn't this be ±3%?

*Yes, this makes more sense and has been changed.*

L59 Change "has" to "was"

*Done.*

L63 Change "representing" to "represents"

*Done.*

L72 "...in part to be compliant with the analysis in earlier inventors." Do you mean communicative? What are the other parts?
*In part means that this is only one reason for incompliance. Apart from the mentioned requirement to be compliant with the interpretation of earlier national inventories, also the different interpretation due to the higher resolution sensor is a major issue. These are actually also the reasons for the missing detailed change assessment.*

L90 What does "commissioning phase" mean?
*It is related to the 3-6 month testing phase of the satellite with all its original settings. Image acquisition during this time is not performed according to the later nominal schedule.*

L94 "...and distribute the raw outlines to the national experts for edition of wrongly classified regions" This sounds smart to use local knowledge, but to me it does not fit the description "consistent". To me a consistent method would be done by one person or one automated routine.
*Yes, fully agreed. As mentioned above, we here refer to consistent on the national level. On the alpine-wide level this inventory is not consistent. However, we had several iterations of some outlines among the participating analysts that helped clarifying some major differences.*

L96-97 "As a guide for the interpretation the analysts used the latest high-resolution inventory in each country" What does this mean exactly? Is the time stamp then some mix of 2015 and the range spanning the imagery used for the national inventories?
*No, it just means that the outlines of the previous inventory have been used as a visual guide for the interpretation of debris-covered glacier parts and possible snow fields etc.*

**Study Region**
L121 "With a total area of about 2000 km^2 in 2003" You say 2100 km^2 on L30 in the Abstract.
*Yes, indeed. The value had been rounded to provide an easy to understand order of magnitude value to the overall mass loss (also the 1 m w.e. per year specific mass loss is rounded). We have clarified this in the revised ms.*

L121-122 Please add a citation for "about 1 m w.e. . . . . 2 Gt of ice per year."
*Done (Zemp et al. 2015).*

L126 "and the mean elevation is around 3000 m a.s.l., a unique value compared to other regions of the RGI." In which direction?
*Both, up and down. It is the only region with the peak area distribution around 3000 m.*

L130 ". . .many glaciers – large and small – become invisible under increasing amounts of debris" "Invisible" is imprecise language, "indistinguishable from optical data" is better.
*It is not only from optical data, it also refers to field observations and geophysical methods. It can simply no longer be decided where a glacier is ending as everything might dissolve into a continuous ice-debris landform (e.g. ground ice, ice-cored moraines, rock glacier, peri-glacial forms). We have changed invisible to hidden.*

L132 "...mapping their extent is increasingly challenging" Is this the same glaciers becoming more challenging or new glaciers becoming challenging?
*Both, the original ones and possible new ones. It just becomes an increasing mess.*

**Datasets**
L149-150 "only the required bands, no longer possible" I don't understand this.
*The policy of distributing Sentinel-2 scenes has changed, so that individual image bands could no longer be downloaded from this source. However, as we have done so, we think it is important to be precise here.*

L161-164 I really like this level of detail you provide. Here you make clearer what "commissioning phase" means (question to L90).
*Thank you. We thought it would be good to already mention it shortly in the introduction.*

L193 Change "using" to "to use"
*Done.*

L194-195 ". . .due to the locally poor geolocation of the S2 scenes . . . the location of the ice divides was [change to: were] partly manually adjusted" Are you saying you moved correctly geolocated flow divided to agree with an erroneously geolocated S2 image? Please be clear what this implies about your new dataset (including the magnitude of these adjustments) and how you motivated your choices.
*Yes, this was required but only locally, i.e. for a few dozen glaciers. The shifts imply that the new divides can only be applied to these Sentinel-2 scenes. Apart from this, it guarantees consistency with the glacier divides used in the previous inventories. This information has now been added.*

**Methods**
L201-207 This is confusing to follow.
*We agree and have rewritten it. It is now less condensed but hopefully clearer.*

L208 "We followed the recommendation to select the threshold in a way that good mapping results in regions with shadow are achieved." Where is this recommendation? Can you please add the threshold values used to Table 1 and cite that here?
*It has been written in several publications. We have now cited Paul et al. (2015) as a more recent study. Theoretically we can provide most of the threshold values used, but practically this would be of limited help as digital numbers were up-scaled from 12 to 16 bit afterwards and a different and more complex pre-processing had been applied to glaciers in Austria. The effect of different threshold values (for th1 and th2) on the classification is illustrated in Fig. 3 and can be used as a guide for other regions.*

L213 You mention several times "misclassified rock in shadow" which I believe means the dark rock becomes darker but is then classified as glacier, I'm confused by this.
*It means that the rock in shadow is classified as ice because its value in the ratio image is the same as for dark glacier ice in shadow. This can happen at the edge of spectral separability when the SWIR band reflectance is becoming very small. We have added this for clarification.*

L222 "contrast stretched" Is this just referring to how you assigned display colors? Probably unneeded information if so.
*Yes, it means we have increased the brightness of the image for better display. Although a minor detail, we think this information is at least relevant for correctness.*

L235 "All pre-processed scenes were provided in their original geometry for correction" This doesn't make sense to me. What is the geometry of a pre-processed satellite image?

*The Alps cover three UTM zones (31-33) and we have not re-projected them to a common geometry (UTM32) beforehand but provided them as is. The pre-processing only refers to the threshold selection for best mapping results.*

L238 Dark bare ice (that you reference in the sentence before) is also an omission error
*Yes, this is correct. We have added 'dark bare ice in shadow' to omission errors and written more precisely 'rock in shadow' for commission errors.*

L239 And shadowed rock classified as glacier is also a commission error. Meaning, you're not wrong but it's slightly odd to not give these aforementioned quantities the same classification.
*Yes, see above.*

L240 I know this is a glacier definition interpretation question, but I really cannot imagine a 0.01 km^2 "glacier" internally deforming. The argument for including these patches is weak, in my opinion.
*Fully agreed, but it stems from the original UNESCO classification for the world glacier inventory (WGI) that also included' perennial snow and ice', i.e. what we would call firn but might not densify to (and flow like) a glacier. The reason for including such features was related to the hydrological background of the WGI, i.e. it should contain all 'frozen water bodies on land'. As in steep terrain such small entities can indeed be 'real' glaciers (i.e. flowing and have a mass balance), we keep them here. For the wider discussion of this somewhat delicate topic we have now added the recent study by Leigh et al. (2019).*

L246 "(see Paul et al. 2016)" do you mean "following Paul et al."?
*Yes, we have now written '(cf. Paul et al. 2016)'.*

L253 Regarding sub meter resolution data to inform debris cover delineation, I agree this can be very helpful, but I think it can also be less helpful. In some instances it is simply not clear, even if you were standing in the field, in these cases I think the high resolution images give us analysts a false sense of confidence. I would even argue that in some cases a lower 10-15 m resolution helps flow features stand out which is probably a better guide to finding the glacier margin than images that can resolve individual clasts. You make a similar counter argument to mine later at L321-323.
*We fully agree but think that it really depends on the specific glaciers. Sometimes images with 10 m resolution provide the required details to identify such regions, and sometimes there is no benefit of the improved spatial resolution. The general issue that one can see different things at different image resolution is a general problem of more recent inventories and also a reason why the comparison to the 30 m resolution inventory from 2003 is problematic. Figure 4d illustrates a part of this interpretation variability for a difficult case.*

L254 "we illustrate the strong glacier shrinkage from 1998" Strong relative to what? And I believe all citations so far have been to the 2003 former outlines, please add a citation for these 1998 outlines.
*The strong shrinkage is from 1998 to 2016. The citation for the 1998 dataset is given in L249.*

L265 "changes. . .are important." Hard to constrain importance, I would say "notable" or "visible"
*Agreed and changed to 'visible'.*

L270-284 The relevance of these two paragraphs is not clear to me. What information does your reader need from this to understand your study? I think it can be said much more concisely.

*This is basically a description of the deviating glacier mapping conditions in Italy and how the additional scene selection has been handled. We think this is required to be transparent. However, we agree that the two sections can be shortened and have condensed the two sections to one.*

L289 Can you please let us know how common these shadow errors were? In km^2 preferably but also qualitative terms could be okay. I think it is useful information for others who will use this work as a guide for their own glacier inventories.

*The regions in shadow have not been measured explicitly but this replacement only affects a couple (31) of very small glaciers (the largest is 0.13 $km^2$) existing in topographically favourable conditions (collectively they cover only 1.35 $km^2$). They do thus only show very small changes over decades and the 'old' outlines are likely more precise than digitizing something that is barely visible on the satellite image. We have added this explanation. We also cite now the study by Fischer et al. (2014), who have investigated the benefits of very high-resolution imagery for glacier mapping when it comes to such small entities.*

L293 Do you mean "[Italian] alps"? also I don't think "i.e." is correct if you list the full set of 3.

*Yes, corrected to 'Italian Alps'. When 'i.e.' means 'that is' (?) it is correct. Using e.g. ('for example') would be wrong.*

L295 If you go into sub-region detail it would be helpful to have these regions labeled in Figure 1 and possibly summarized in a table.

*We have now added the respective tile number (14) from Fig. 1. We provide some more details for Italy as the region is rather extended and scene selection was more complex than for the other three countries.*

L300-304 I think this is an appropriate place to mention rock glaciers and either cite others regarding their definition/interpretation or use these data and possibly change since 2003 to make your own inference on this classification. Were you sufficiently convinced what you show in Figure 4c is in fact glacier? You defer to the previous inventory, but you clearly demonstrate a level of expertise in this subject and I think the readers will appreciate a more decisive call on these small, difficult glaciers.

*The shown close-up is indeed a region on the edge. Whereas sharp ridges indicate that there should have been a glacier before, what remained in 2007 and also 2016 might no longer be classified as a glacier. The satellite image of the two larger glaciers (nr. 529 and 530 in the last Italian inventory) basically shows a debris-covered slope whereas the aerial image shows a trough filled with avalanche snow (seasonal snow as mentioned in the text). There might still be ice underneath, but calling this a glacier would likely only be justified when knowing that there has been a glacier before. For this inventory it is still included, but on page 179 of the book in the last Italian inventory (see http://sites.unimi.it/glaciol/wp-content/uploads/2019/02/5-Lombardia.pdf) it is written for both glaciers 'Almost totally debris-covered; in the CGI inventory it is labelled as "extinct"'. Hence, this is a very difficult case. We have now cited the study by Leigh et al. (2019) and added a short comment.*

L317 I think something like "completely ablated" is more precise than "disappeared" but maybe only stylistic.

*See above, disappeared should also include that they might still be there but that they are out of sight (e.g. buried under debris from rock-fall). In this case 'completely ablated' would be misleading, in particular when we do not know what has happened process-wise.*

L325-326 "This glacier has thus strongly grown since 2003 due to a new interpretation..." This is incorrect language, change to "the interpreted glacier area strongly grew"
*Fully agreed, it is not the glacier that has grown but the delineated extent (changed).*

L336-339 I agree with your statement that political boundaries are meaningless in a scientific context, yet you go on to present your results per country. I understand that the source of some degree of your results are from national inventories, but my sense was this article is a pure research stitching together of these datasets. What is your motivation to partition your results by country?
*It is basically only because we asked the national experts to do this. Despite the tough work, the work was not specially funded for some countries but cross-funded from other projects. Some national results thus help to highlight the contribution of the partners.*

L341 I don't think "digital" is needed.
*We agree that the statement would also work without the 'digital', but think that not all readers are familiar with GIS-based data processing. So for some readers this might be relevant information and we would thus like to keep it.*

L360-362 Please provide the specific rational as to why a glacier specific comparison was not possible between glaciers that met the "point in polygon" check. One thing that makes this article so unique is that the 2003 inventory was made by you. This is the interesting point of unique consistency that you bring with this study but here it seems like you deflect from this. Has your personal definition of a glacier changed so much since the earlier inventory? There is a strange human element to this line of work/this study, which you later spend considerable effort attempting to constrain, yet this change in interpretation/definition of a glacier between the 2003 inventory and this one, with the same lead author, is surprising and interesting to me.
*This might indeed come across a little bit strange, but there are a couple of reasons for it:*
*1. The interpretation is indeed partly different due to the better resolution of Sentinel-2. This led to small changes for some glaciers that partly resulted in a change of topology (e.g. two units are now connected or have split into several parts). Such changes can only be considered manually and this would be a considerable effort.*
*2. The 2003 inventory was also created by three different persons that have - despite all cross-checks - slightly different interpretations. It would be possible to directly compare extents on a glacier-by-glacier basis for the part that the first author has done, but this would only be a subsample of all glaciers and still suffer from point 1.*
*3. For Austria, the differences in interpretation are very large (see Fig. 10) and a direct comparison with the 2003 outlines thus not possible. The differences in Austria are mainly caused by (i) the improved spatial resolution of the Sentinel-2 images and (ii) by considering the previous national inventories generated from orthophotos and lidar data (Lambrecht and Kuhn 2007, Fischer et al. 2015) for consistency checks with the new inventory, whereas the 2003 outlines were derived from 30 m Landsat images without using existing datasets for consistency checks.*

L382 ". . .we applied the buffer method. . ." Please add a citation for this
*Added (Paul et al. 2017)*

L390 I don't understand how it is computationally expensive to buffer the debris-covered areas (a smaller area than total glacier area) while it is computationally feasible to buffer all areas. I have applied buffers to all glaciers and debris cover on Earth on a typical laptop without outstanding computational demand. Is there a step or condition that causes the high computational cost?

*There are at least three reasons for it (cf. Mölg et al. 2018): First, we do not get the debris-covered regions automatically (e.g. from a subtraction of the clean ice part from the corrected extents) as we have also corrected other regions manually (e.g. shadow, supra-glacial lakes). Second, application of the buffer method to a given set of outlines (polygons) is a very quick thing, fully agreed. But for the uncertainty assessment it is required to only use the outer perimeter of the debris (as it does not matter for debris enclosed by ice) and identifying this part of the perimeter is quite an effort as this is in general numerous bits and pieces rather than one closed polygon. Finally, there are different types of debris covered glacier areas. In some cases the debris boundary is clearly defined and can be accurately traced, sometimes it is rather a guess based on the analyst's knowledge (see Figs.4d and 11). Hence, we have decided to not perform this (statistical) assessment here but focussed on the multiple digitizing.*

L391-392 I think Figure 4d is a good example of where this assumed realistic uncertainty estimate of ±2 pixels is not applicable. You say this depends on the degree of debris coverage which I agree with, could you apply a different buffer as a function of debris-covered area? Maybe it could help keep your above referenced computational cost low if only considering the very debris-covered glaciers.

*Thank you for this suggestion. As mentioned above, apart from the degree of debris cover it is also the topological complexity of the locations with debris cover that increases computational cost. We have also no automated method to determine the 'very debris-covered glaciers'. The glacier to the NE of Glatscher da Gavirolas (named Glatscher da Fluaz) in Fig. 4d is also very challenging to digitize and can be about 100% larger than in our interpretation as the valley floor is to a considerable extent filled with debris-covered ice. In other words, the result obtained with a more complex buffer method (giving an uncertainty between 5 and 10%) would not reflect the real problem. We have now better described these uncertainties.*

L396-397 Please watch your significant figures, the sum your report is slightly off. Further, do you have confidence in your results in Austria to 0.01 km$^2$?

*The difference is due to the rounding of values, it is just 0.1 km$^2$. As discussed above, the 0.01 km$^2$ threshold is certainly the lowest sensible size for a glacier. This value is also valid for Austria, but here some entities being much larger (say 0.1 km$^2$) are included where we can certainly discuss if these are glaciers or perennial snow/ice features (e.g. persistent avalanche deposits). So it is less the minimum size that is problematic and more the different rules applied in each country to include certain features or not.*

L399-402 I do not see the value in 'number of glaciers' based results. I am not convinced a different research group repeating your study will fine the same number of glaciers. If you disagree with this argument, I would at least suggest a minimum area of 1 km^2 to promote some degree of repeatability.

*Even at that size it will likely be difficult due to some topological differences (see above L360). However, all values can be taken from Table 2. Apart from this, we 100% agree that the number of glaciers is science-wise the least relevant and has a very limited meaning. On the other hand, it is the most reported (e.g. the 215 000 glaciers in the RGI are widely known, but the total area?). So counting entities is highly important for communication, even if the number makes very little sense.*

L415 "glaciers smaller than 1 km^2 can be found at all elevations" I see what you are trying to say but this statement is incorrect.

*Do you mean that these smaller glaciers cover 'nearly the full range of possible elevations' rather than 'all elevation'? This is fully agreed and changed accordingly.*

L415-416 "indicating that their mean elevation [remove: does] only slightly depend [depends] on climate factors" This doesn't make sense to me. I think you are trying to obliquely address the question: why aren't some of these small glaciers bigger? I think there might be a statement you can make here but as it is I don't think there is supporting evidence.

*It simply means when small (and thus thin) glaciers are also found at low elevations (where temperatures are higher), they must be protected from the direct climate forcing (e.g. due to shadow, debris, avalanches, see above). Or in other words, their existence only slightly depends on climate factors (still they need some snow for nourishment). We can discuss if these entities can be called glaciers (e.g. do they flow or is it just a perennial snow/firn accumulation), but in most cases field investigations might be required to decide it (so for the time being we keep them included). As reviewer #1 had a similar comment, we have added a statement for clarification.*

L417 "arrange around a climatically driven mean elevation which is around 3000 m a.s.l." Why not compute the mean and add a fit mean line to Figure 6a? "...the largest glaciers are not those with the highest elevation range. . ." Yes, I actually think they are and your next statement contradict your previous statement ". . .and for the majority of glaciers the elevation range increases with glacier size" One very steep, small glacier is an outlier, not relation breaking evidence.

*First point: When mean elevation is plotted against area this would basically give a straight line as values are normally distributed around the mean. So this does not make for an interesting plot and can also be described by a short text.*
*Second point: There are at least five glaciers with a higher elevation range than three larger ones. In this case it is a characteristic of the Mt. Blanc mountain range that is very high and very steep and does not allow building large glaciers. We would not mark it as an outlier as it applies to several glaciers and as the Mt. Blanc region is a regular part of the Alps.*

L422 "This is typical for regions dominated by mountain and valley glaciers as these follow the given topography" This statement needs a citation and an example of regions where glaciers (or ice sheets) do not follow the given topography.

*As mentioned in the following example, Plaine Morte glacier is an exception from the rule as it is a flat plateau glacier. This type and ice caps / ice fields do not follow this rule. We are not sure if there is a citation for this as it just follows from the geometry.*

L423-424 "an exception from the rule" What rule? I would call it an outlier

*The rule would be: The larger the glacier, the larger its elevation range, and we have now added this for clarification. This rule only applies to mountain and valley glaciers rather than plateau glaciers (or ice caps / ice fields).*

L428 Can you please motivate your use of median here versus mean at L415.

*There is no physically based motivation as all three elevations (mean, median und mid-point) carry about the same information. The non-physical motivation is that median elevations have been used widely to display the spatial variability.*

L428 "largely driven by temperature, precipitation and radiation" Since you don't specify glacier size this is in contradiction to L415-416.

*We agree that this seems to be a contradiction at first glance, but the statement here refers to 'usual' glaciers rather than those existing under special conditions. The latter results in small glaciers at low elevations so that their (mean or median) elevations are also about normally distributed. There is thus no dependence of (mean or median) elevations on glacier size. To remove the large scatter of the small glaciers normal distribution and see the relevant signal, the map in Fig. 7 only shows glaciers larger than 0.5 km².*

L428 Wouldn't topography be an important factor here?
*Topography affects mass balance (and thus location) basically via the mentioned radiation receipt and (possibly increased) snowfall amounts. So topography is implicitly considered. However, to be clear we have now also added topography (in brackets).*

L429 "temperature is rather similar at the same elevation over large regions" Needs citation
*We hoped this is common knowledge but found a study that has now been cited (Zemp et al. 2007).*

L431 Remove "amounts"
*Done.*

L434-435 Why are glaciers larger than 0.5 km^2 les impacted by local topographic conditions?
*The 0.5 km² threshold is arbitrary and has been selected to create a good display of all values in Fig. 7 (the inset shows the full sample). Otherwise, the portion of impacts from topography (e.g. shadow) decreases with increasing glacier size (for mountain and valley glaciers, not valid for ice caps).*

L435 ". . .median elevations (around 2400 m a.s.l.)" L418 says 3000 m a.s.l.
*We here refer to approximated minimum and maximum median elevations related to their dependence on location (Fig. 7) and mean aspect (Fig. 7, inset) rather than to the mean value for all glaciers.*

L438 "b" not labeled in Figure 7
*Indeed, thank you for spotting (changed now to Fig. 7, inset).*

L439 "On average, glaciers. . .have median elevations that are about 400 m higher" Is this a mean of medians? I'm a little lost how you choose distribution middle metrics.
*There are indeed quite a lot. Here we have not calculated mean values per aspect sector but just visually averaged the point clouds depicted in the inset plot of Fig. 7.*

L440 "However, the scatter is high" It's hard to understand a "result" that is then walked back some qualitative amount. Can you draw on statistical tests to qualitatively inform signal form noise?
*Also this value is only derived from a visual analysis of the scatter plot. There should not be a signal from noise separation as both values are equally relevant without impacting on each other. Indeed, the scatter plot is just reporting that there is some impact of radiation receipt left, but there is also a high variability for each aspect direction as spatially highly variable precipitation amounts exert a strong influence on absolute values. As a back-up for the visual analysis we have now calculated aspect-sector averages and standard deviations of mean elevation and added the related values to the plot.*

L445: I am again missing the scientific argument for presenting result per country.
*There is no scientific argument for it, it is more a political decision that 'comes with the territory' when national experts are contributing.*

L456 "For a selection of 2873 glaciers present in both inventories. . ." According to this, the number of glaciers in 2003 was 65% of the number of glaciers now. I find this more concerning than a result. I am inclined to assume this new inventory is overly generous with a "glacier" classification and I would strongly encourage the authors to double check their confidence on what they are calling glaciers.

*As mentioned above, people (incl. reviewers) are highly interested in the number of glaciers, even if this is a largely meaningless value ;) We have now written 'for 2873 comparable polygon entities' instead of 'glaciers present in both inventories' as this has also a topological component. For the most part, the excluded glaciers are very small and relate to not mapped entities in NE Italy (in 2003) and partly much larger glaciers in Austria due to another definition of glaciers (see Fig. 10). We have accepted this deviation for this study to include the work of the Austrian colleagues, but otherwise we agree that it also creates some trouble when going into details.*

L460-461 Per country results should probably be in a table if you report some here.

*This could be done but it would be biased by the different size-class distribution in each country as relative area changes have a size dependence. Strictly, it would be required to do this size-class dependent and this is somewhat beyond the scope of this inventory presentation (and would require a considerable extra effort as glaciers split and change size classes etc.). So if you don't mind we would prefer going ahead with the aggregate numbers.*

L462 "Reveals [should be: a] small shift" Can you please say more about this translation error, it doesn't look linear(?) is it elsewhere? Should this be corrected?

*It was a matter of big discussion with space agencies as the 90 m DEM they used for orthorectification of the Sentinel-2 scenes was not accurate enough, creating irregular pixel displacements. Additional, the set of ground control points (GCPs) used had some issues resulting in a small (about 30 m) more systematic shift. This caused the problems with applying the drainage divides used for the 2003 inventory also to the Sentinel-2 scenes. As far as we know, the problem with the DEM has been solved in the meantime but we had to use the images from the commissioning phase at that time.*

L466-468 As stated above I'm concerned there is not a way to make a per glacier area change plot. In my opinion, the new outlines (yellow) in Figure 10 have a lot of unrealistic area: too narrow, too small, not following a flow pattern. For the glaciers that do intersect 2003 it is clear the outlines are of very high quality, but I think the 2003 interpretation of what is a glacier is in some cases better.

*We agree to the latter but as explained above the Austrian inventory had included these 'perennial' features for consistency with earlier inventories. Most of them can be seen as water resources and have thus a right to be included, but many of them might not be glaciers in a classical sense (with flow, accumulation/ablation etc.). On the other hand, when carefully analysing the very high-resolution image available in Google Earth (from 13. Aug. 2015) for the region shown in Fig. 10 one can also argue that many of these features indeed consist of glacier ice and have been missed in the 2003 inventory. One can thus also argue that the 2003 inventory underestimated glacier area rather than the new one overestimating it. Both have their shortcomings (as illustrated in L476ff) and a related scatterplot would be dominated by the interpretation differences rather than real area changes. We think this is not meaningful. So apart from the differences in interpretation, here is also a warning that inventories that have been derived from sensors of different resolution can be incomparable. This problem has also been raised by Fisher et al. (2014) and Leigh et al. (2019) that we have added here. In short: Instead of forcing the comparison to work, we would prefer stating here that it can not work.*

L478 "with a limited meaning on the basis of individual glaciers" These differences that disable a per glacier change analysis aggregate to the whole, I'm not sure how you can consider changes of the whole Alps or per country and not be able to consider individual glaciers.

*As mentioned above, in part the per-glacier assessment has not been performed because automated methods cannot be applied due to the topological changes and the fact that ID assignments had to be performed manually. These difficulties can be avoided when only referring to aggregate values such as for Fig. 8 or the total. However, there is likely no error compensation but a 'too small' bias for 2003 and a 'too large' bias for 2015/16. Considering these biases, the relative area change rates will increase (see next point).*

L480 "-15%" This value starts at -13.2% earlier in the manuscript, was bumped up to 15% at L459 and now is "even higher". For the results section I think it's best to present a consistent and confident value.

*The -13.2% is only referring to the subsample of 2873 glaciers, the other values given are extrapolations that consider possible biases in the dataset (see point before). At first we consider the underestimation of area in the 2003 dataset (e.g. missing debris covered parts and missed small glacier), giving a larger 2003 area and thus a higher loss. In the next step we consider that with a different interpretation (e.g. perennial snow and ice excluded) the 2015/16 extent would be even smaller. This would further increase the area loss (exceeding 15%). So this is just an attempt to compensate for known biases in both datasets to present a more realistic value.*

L493 "a detailed analysis" Where is this analysis? Or do you mean the overlay Figure only?
*It just refers to Fig. 11. In the GIS we have zoomed in to the individual regions (taking also some of the outlines off) to better see where the differences in interpretation are located. All of these regions are visible in Fig. 11.*

L495 "When excluding P1" what if P1 is the most trained expert?
*Than all other participants are too conservative in their interpretation and P1 has to explain why the additionally considered regions under debris cover should belong to the glacier. But the text as written would still be correct.*

L505 "digitized glacier extents increased by several per cent after consultation of very high-resolution satellite images" Is there a way this information could be added to Table 3?
*This would basically be four additional tables showing lots of numbers. We can add them in a supplement but think they do not add much. Showing how the outlines have changed after the consultation would likely be more relevant. We have prepared a couple of examples for a supplement illustrating the change in interpretation and some critical regions at very high resolution.*

L510-513 "If such regions have to be included...can be discussed...including these features in a glacier inventory or not is a (personal) methodological decision" This belongs in the discussion section.
*We agree, but think it can also be mentioned here as otherwise we need to repeat the entire context for this statement in the discussion before making it.*

L522-524 Check English here.
*The sentence has been revised. (Now: "The comparison of topographic parameters (minimum, maximum and mean elevation, mean slope and aspect) as derived from the TDX and AW3D30 DEM revealed larger differences, in particular towards smaller glaciers.")*

L528 re-state acquisition gap dates.
*Added.*

L530 5 years of elevation change are within the uncertainty range? I would expect there to be a clear signal.
*We have not corrected TDX for radar penetration and do not know the exact years of image acquisition as both datasets are merged products from several years. If the scenes over the Alps are from 2010/11 for AW3D30 and 2012/13 for TDX, we do not expect to see a clear elevation change signal.*

**Discussion**
L535 missing citation here
*We cite now Pfeffer et al. (2014) who show related histograms.*

L536-537 Probably belongs in results section
*We agree but think this is only data already presented in the results (though slightly differently aggregated).*

L538-539 "However, for consistency with earlier inventories they have been included" Figure 10 makes it clear this is not true, consistency with earlier inventories would exclude non-glacier area. I think it's concerning that the authors walk back what is considered a glacier here in the discussion, in my opinion, we need this set of experts to make these (hard) decisions so the rest of the community can uses this product with confidence.
*To the first point: We have changed the sentence to "with earlier national inventories" to be clear we are here not referring to the 2003 dataset.*
*To the second point: Fully agreed, it would be good when we can reach consistency here. The currently best suggestion we have on the table is that analysts can include in their (national) inventories whatever they want, but there should be a mark in the attribute table what the polygon is describing (e.g. glacier, perennial snow/firn, rock glacier, regenerated glacier, includes dead ice, etc.). This would allow the user of the datasets to later select them as required for a specific application. We are not yet there but something like this will hopefully be established soon.*

L540 "precipitation amounts have a limited impact" I think this a contradiction to L431.
*In L431 we talk about the impact of precipitation amounts on glacier location for those glaciers that do not benefit from special topographic conditions (e.g. the larger ones). Here we talk about the other 'glaciers', those found in topographically preferred locations that can be found at any elevation. There is thus only a limited impact of precipitation amounts.*

L543-544 I don't think Figure 7 supports these claims.
*The Fig. 7 inset does indeed not show glacier size but the map has been restricted to the larger glaciers. So 'modified by glacier location with respect to precipitation sources' is correct and 'modified by mean aspect' is also correct. The sentence has been rewritten to disentangle both observations. (Now: "Glacier mean elevation does not depend on glacier size but on glacier location with respect to precipitation sources, in particular for larger glaciers (Fig. 7). On top of this dependence is the variability with mean aspect (Fig. 7, inset).")*

L546 "Widespread glacier thinning" L530 suggests no thinning
*We here refer to a much longer time period (last 2-3 decades) than for the DEM comparison (2-5 years). (Now: "Widespread glacier thinning over the past decades and …")*

L546-547 confusing wording
*Rewritten in two sentences to be clearer.*

L548 Confusing English in this sentence
*Rewritten. (Now: "These separated parts can thus not be named 'regenerated glaciers' but they melt away as dead ice.")*

L551-552 "merged their IDs. . .combined extent" This is unclear
*Despite being two separated polygons, both have a common ID so that all topographic information is calculated for the combined ice masses rather than the individual polygons. (Now: … and used the same ID for both parts to obtain topographic information for the combined extent.")*

L558-559 As stated above, interpretation/definition errors don't disappear at a wider scale.
*Yes, this is correct but it allows automated computation as topology issues can be neglected. For this reason we have added the two further extrapolations that consider the interpretation biases (possibly too small/large glaciers in 2003/2016, respectively).*

L565 Snow covering 20% to 30%, where did these numbers come from, I don't recall anything in the methods.
*This has not been explicitly calculated but can be seen on the cited Figs. 9 and 11 (i.e. it is only a qualitative estimate).*

L571 "change rates of identical size classes are compared" Do you mean hypsometry?
*No, mean area change rates per size class (e.g. as shown in Table 2 of Paul et al. 2004).*

L573 "revealed a large variability in the interpretation of debris-coved glaciers" Did it? I'm not sure if this was quantified.
*It can be seen in Fig.11: All regions where the differently coloured lines are not on top of each other (so that only one colour is visible) are debris-covered glaciers. The larger the spread, the more difficult the interpretation has been.*

**Conclusions**
L609 "DEM quality and co-registration" were these mentioned in the text?
*Indirectly. In L246 we refer to the further processing that is described in detail by Paul et al. (2016).*

**Tables and figures**
Table 3
Am I reading correctly that STD derived for n=4? Is glacier ID 4 two glaciers in the inventory? This sounds like an interesting result that most participants called in one. Is this a unique case, or common?
*First point: Yes, the STD is for n=4.*
*Second point: For the sample of glaciers we had selected it was only this one. The split into two glaciers had been made based on the Sentinel-2 image only. During the digitizing and checking back with the higher resolution imagery it was revealed that both glaciers are still connected under debris cover. So when two parts had been digitized, both parts have been summed up.*

Figure 3
The blue is a little misleading, possibly better as 4 sets of colored lines It's a little hard to tell what is error and what is true.

*We can exchange the blue with another colour but actually the colours only indicate what is mapped with the related thresholds. There is yet no true or fase included. The blue colour is depicting (in b and c) the most conservative mapping. When lowering the respective thresholds, the yellow, red and green parts are additionally mapped. Panel d shows the consequence of the selected threshold th2=860, correct mapping of rock outcrops at the white arrow, and at the same time missed ice in shadow at the green arrow.*

Figure 4
Are blue in a) and green in b) the same? Why different colors?
*If blue in a) means light grey, yes these are the raw mapping results for both regions. Colours are different because analysts have chosen the examples for visualization independently. Not perfect but maybe acceptable.*

Figure 9 and 10 Many of the small glaciers identified in these figures do not look like glaciers to me.
*Fully agreed, at this scale they are hard to identify and some of the extensions might indeed only be perennial snow/firn. However, please check for both also the very high-resolution imagery available in Google Earth. Suddenly real glaciers materialize ...*

---

## Author Response (AR1)

[revised manuscript text omitted]

The comparison of glacier outlines in Fig. 10 illustrate for the region around Sonnblickkees in Austria why we do not provide a scatterplot of relative area changes *vs.* glacier size or country specific area change values (cf. also Fig. 4d for Gavirolas Glacier in Switzerland). Due to the different interpretations in the new inventory, 125 mostly very small glaciers are 100% to 630% larger than in 2003 and a large number (557) is 0% to 100% larger. For example, the 4 km$^2$ Suldenferner has increased in size by 550% as a small tributary (that holds the ID for the glacier) was disconnected in 2003 but is now connected to the entire glacier. Although such cases can be manually adjusted, it would not solve the general problem of the different interpretation when using data sources with differing spatial resolution (cf. Fischer et al. 2014, Leigh et al. 2019). For example, the glacier in Fig. 4d has increased its size from 2003 to 2015 by 56% due to the new interpretation. On the other hand, Careser glacier, which fragmented in six ice bodies from 2003 to 2015, lost 55% of its area when summing up all parts as opposed to 63% when considering the largest glacier only. In consequence, the possible area reduction due to melting is partly compensated by the more generous interpretation of glacier extents and thus with a limited meaning on the basis of individual glaciers. Overall, glacier extents in the 2015/16 inventory might be somewhat larger than in reality due to the inclusion of seasonal/perennial snow in some regions. The -15% area loss mentioned above can thus be seen as a lower bound estimate.

*Figure 10*

**5.3 Uncertainties**

**5.3.1 Glacier outlines**

The multiple digitising experiment revealed several interesting albeit well-known results. Overall, the area uncertainty (one standard deviation, STD) is 3.3% across all participants for the total of

fp 28 4 20 3:19 PM

fp 28 4 20 3:19 PM

fp 28 4 20 3:19 PM

fp 28 4 20 3:19 PM

fp 28 4 20 3:19 PM

fp 28 4 20 3:19 PM

fp 28 4 20 3:19 PM

fp 28 4 20 3:19 PM

[revised manuscript text omitted]

) b)

**Figures**

[Figure]

Figure 1

[Figure]

Figure 2

fp 28 4 20 3:19 PM

fp 28 4 20 3:19 PM

[Figure]

Figure 3

[Figure]

Figure 4

b)

[Figure]

Figure 5

fp 28 4 20 3:19 PM

[Figure]

Figure 6:

fp 28 4 20 3:19 PM

[Figure]

Figure 7

fp 28 4 20 3:19 PM

[Figure]

Figure 8

[Figure]

Figure 9

km

[Figure]

Figure 10

fp 28 4 20 3:19 PM

[Figure]

Figure 11

fp 28 4 20 3:19 PM

---

## Author Response (AR2)

**Response to the comments by the Editor**

Dear Authors,
Thank you for submitting your revisions and for responding to the reviews. Both reviewers have clearly indicated that this is a useful and comprehensive dataset worthy of publication. I do agree with the reviewer's recommendation and congratulate all of you for an extensive and meaningful piece of work.
*Thank you*

Below I have a number of minor remarks that should be considered prior to moving on to the next step.
*Thank you for the careful reading!*

l. 33 increased in size (small glaciers)
*Rewritten to: '... many small glaciers were additionally mapped or* ***they*** *increased in size compared to 2003' to be clear that the second part of the sentence is also referring to the small glaciers.*

l. 40 Remove "Precise"
*Done.*

l. 64 "very high-resolution" -> "better resolved"
*Done.*

l. 89 remove "In this study,.."
*Rewritten as 'We here present ...'*

l. 91 "in in"
*Second 'in' removed.*

l. 99 specify number of participants
*Added (five).*

l. 138 remove "unique" (unless there are non-unique tiles which would require an explanation).
*The point is that some tiles have scenes from different dates so the number of tile IDs is smaller than the number of scenes processed. To reflect this we have now written 'We processed 17 different S2 tiles from a total of eight different dates ...'*

l. 184 remove "As mentioned above," and start with "Outlines…"
*Done.*

l. 186 "with very high spatial resolution (better than 1 m)" -> "with a spatial resolution smaller than 1 m"…
*We think the more widely used terminology is 'better than'. We have thus now written: 'with a spatial resolution better than 1 m'*

l. 189 Quantify "very small"
*We have not quantified this explicitly and it is actually quite a range as some 'larger' glaciers (debris covered) are also in the mix. Some of them might even not be considered as 'real' glaciers as the comments from reviewer 2 revealed. We thus think providing an explicit threshold here is misleading and would prefer to stay qualitative here.*

l. 190 I don't understand "..digitised with larger extents.". If they weren't mapped before, how can their extents be "larger" (compared to what?)? Rephrase.
*Agreed and rephrased.*

l. 196 Quantify or estimate "locally poor geolocation" in meters.
*We have not measured this explicitly. It is locally variable and in the range of 1-2 Landsat TM pixels. We have added now 'by up to 50 m'*

l. 201 "straight forward" -> "straightforward"
*Done.*

l. 203 Abbreviation NDSI occurs only once. Remove.
*This is correct but we prefer to keep it as the abbreviation is likely more widely known than the full term (scientists either use the band ratio or the NDSI method for glacier mapping).*

l. 227 How do you know that the rock shadow is wrongly mapped? Supposedly because you know the test site very well, but this should be stated explicitly. The same holds for the following sentences. It is unclear where the knowledge of "correctness" comes from.
*It is certainly a mixture of points such as knowing how a glacier works, being clear where snow accumulation is possible, interpretation of differences in reflectance, comparison with previous interpretations, consultation of higher resolution imagery or scenes with even better snow conditions, etc. Overall it is 'visual inspection' and 'expert judgement' which means it must not be true or correct in an absolute sense, but it is only one possible interpretation. We have added the former point in the method description and the latter in the discussion.*

l. 234 remove "before".
*The before is actually extremely important to get the topology right (i.e. rock outcrops represented as data voids). To be clearer about this, we have now started the sentence with this important aspect (as it is the step before the raster vector conversion).*

l. 241 "very small snow patches" rephrase quantitiatively. Do you mean "snow patches < 0.01 km…" ?
*In this case, yes (added).*

l. 254 is "pixel spacing" something different that the wording "resolution" used earlier? If not, stick to using "resolution" only.
*Strictly speaking, yes as the pixel spacing is a regular raster and resolution can be something different (e.g. details smaller than the pixel size might be resolved if contrast is sufficient). However, here it has the same meaning and has been changed.*

l. 324 again "very high resolution". Too many things are "high-resolution" in this paper. Better state or at least estimate the gridding in meters. Alternatively refer to "higher-resolution" and make sure what the comparison refers to.
*We have indeed made intensive use of this more vague term, but it is on purpose. On the one hand, we refer to 'classic' (?) view that Landsat type sensors are high resolution and MODIS type is moderate resolution. This translates to a 5 or 10 to 100 m pixel spacing for high resolution and 100 to 1 km for moderate resolution. Everything coarser then 1 km is coarse resolution and everything better than about 5 m is very high resolution. A distinction between high and very high resolution is also their price, for the former the raw data are freely available and the latter are not. But you can look at these high-resolution images for free in various map servers. Though being very helpful, quantitative information about the*

*sensor (name, resolution, date) is often unavailable. Which brings us back to the distinction made in the paper: very high-resolution data are those we can look at but not process. In this meaning we have used the terminology. This explanation is likely too long for the paper but by writing now: 'very high-resolution satellite imagery or aerial photography (as available in Google Earth or from map servers)', the distinction between the datasets as described above is hopefully clear.*

l. 471 How large is that shift in meters? Due to wrong geocoding?
*We have not measured it explicitly but it is again about 1-2 TM pixels. We have added '(about up to 50 m).*

l. 479 define the term "very small glacier" once and then use it consistently, or (which is what I would prefer) just state it explicitly (here glaciers < XX km^2 ..)
*This could indeed be helpful but we do not use this term in a strict quantitative sense. Sometimes it means <0.01 $km^2$, sometimes <0.05 or <0.1 or <0.5 $km^2$, partly also depending on the context. In some cases it has also not been explicitly measured and/or it includes deviations. So when we write '125 mostly very small glaciers' the majority is likely smaller than 0.1 $km^2$ but a few can be much larger e.g. 1 $km^2$ or more. Giving an explicit size threshold in such cases would be confusing or misleading. Finally, sometimes (like here) the size does not really matter as it has no impact on anything and is more a personal judgement in a specific context. This is different from values relevant for the dataset, e.g. our minimum size threshold for the inventory that must be given. The study by Leigh et al. (2019) is cited in L243 and provides an overview on these issues and the range of interpretation.*

l. 518 "very high resolution"
*Changed.*

l. 526 "even at this high spatial resolution" I am again lost which resolution you are refer to. Use numbers instead. Also in the following lines.
*Please see the lengthy response above (L324).*

l. 540 Large differences in topographic parameters, or large differences between the two DEMs? Rephrase.
*Done.*

l. 542 remove "see"
*Done.*

l. 548 Add "radar penetration in snow/firn". I don't think there should be radar penetration issues for bare ice in the ablation zone.
*Done.*

l. 558 remove "really"
*Done.*

The entire section 5 contains many discussion elements (e.g. l. 523 "…can be discussed.") and I don't see the section 6 heading "Discussion" as justified. Please make the distinction between the two sections Results and Discussion more prominent by moving content from the Results into the Discussion.
*We agree that Section 5.3 includes elements of discussion but found none in Sections 5.1 and 5.2. We can make the entire Section 5.3 a subsection of the Discussion Chapter but think that this is more a matter of personal taste than a strict requirement. The discussion*

*elements in Section 5.3 are in our opinion rather close to comments on results interpretation rather than a real discussion. Moving them to another place where they are then out of context (i.e. it is required to repeat substantial parts of the context to understand the statements) is in our opinion not very meaningful. Our current discussion section is providing some further context and interpretation to the results achieved and seems thus to be justified.*

Figure 2 label (b) is barely visible. Use a white background box. Are the "cJaxa" and "cDLR" signs really necessary and what is the difference compared to a citation?
*The 'b)' label should now be better visible. The (C) is required for DLR according to the user License (point 6) but not for Jaxa*

Figure 7 Label Elevation needs reference (e.g., a. s. l. or WGS84,…)
*We only provide here the unit of 'Elevation' (m) rather than the reference system it refers to. The datum of the DEM used (WGS1984) is given in Section 3.2 and applies to all graphs/figures referring to 'Elevation'.*

Add spatial coordinates to Figures 2, 3, 4, 9, 10, 11
*This is difficult for some of the images (e.g. Fig. 4) as they are just screenshots from the co-authors or the very high-resolution images from Google Earth or other map servers. As the projection is in UTM32 coordinates, explicit annotation would take a lot of space. However, we agree that some orientation has to be provided and have decided to add a marker (+) on each image and provide its geographic coordinates in the figure caption. We hope this is acceptable.*

Labels on many Figures are too small (e.g., inset Figure 7) or boarderline small. Please make sure to submit publication ready Figures for the final version.
*The size of the inset in Fig. 7 has been increased. Labels for Figs. 5 and 6 can also be increased if required. We can provide the bar charts and scatterplots in vector format (eps or pdf) if resolution should be an issue.*

[revised manuscript text omitted]

fp 27 5 20 3:42 PM

fp 27 5 20 3:42 PM

fp 27 5 20 3:42 PM

fp 27 5 20 3:42 PM

**Figures**

[Figure]

Figure 1

[Figure]

Figure 2

[Figure]

Figure 3

[Figure]

Figure 4

[Figure]

Figure 5

Figure 6:

Figure 7

fp 27 5 20 3:42 PM

fp 27 5 20 3:42 PM

fp 27 5 20 3:42 PM

[Figure]

 Figure 8

[Figure]

 Figure 9

[Figure]

km fp 27 5 20 3:42 PM

Figure 10

fp 27 5 20 3:42 PM

Figure 11

---

## Author Response (AR3)

**Response to the comments by the Editor**

Consider using subscripts for thresholds th_1 and th_2
*Done.*

Figure S1 enter scale bar
*Added.*

Supplemental Data: Is that shp file not part of the doi repository?
*In principle, yes (we have to update the version on Pangaea). After we had received a couple of requests for the dataset but still the embargo on Pangaea, we thought it might be useful to provide the shapefile also as a Supplement. We hope this is ok.*

l. 220 Unify using "Figure" or "Fig."
*We always use 'Figure' at the beginning of a sentence as I have read somewhere that starting a sentence with an acronym/abbreviation or number is bad style ;)*

l. 226 th1 is not fully in italics
*Changed.*

l. 405 Unify use of significant digits (.88 km v.s .1 in line 406).
*Done.*

l. 438 "receipt" I think you mean "received" ?
*'Received' would work as well but I think 'radiation receipt' is the correct technical term.*

l. 604 remove "so-called"
*Done.*

l. 643 please add: "The editor thanks Andrea Fischer and Sam Herreid for constructive and meaningful reviews."
*Done.*